# First high resolution BrO column retrievals from TROPOMI

Sora Seo[1], Andreas Richter[1], Anne-Marlene Blechschmidt[1], Ilias Bougoudis[1] and John Philip Burrows[1]

[1] Institute of Environmental Physics, University of Bremen, Bremen, Germany

*Correspondence to*: Sora Seo (sora.seo@iup.physik.uni-bremen.de)

**Abstract.** For more than two decades, satellite observations from instruments such as GOME, SCIAMACHY, GOME-2 and OMI have been used for the monitoring of bromine monoxide (BrO) distributions on global and regional scales. In October 2017, the TROPOspheric Monitoring Instrument (TROPOMI) was launched onboard the Copernicus Sentinel-5 Precursor platform with the goal of continuous daily global trace gas observations with unprecedented spatial resolution. In this study, sensitivity tests were performed to find an optimal wavelength range for TROPOMI BrO retrievals under various measurement conditions. From these sensitivity tests, a wavelength range for TROPOMI BrO retrievals was determined and global data for April 2018 as well as for several case studies were retrieved. Comparison with GOME-2 and OMI BrO retrievals shows good consistency and low scatter of the columns. The examples of individual TROPOMI overpasses show that due to the better signal to noise ratio and finer spatial resolution of 3.5x7 $km^2$, TROPOMI BrO retrievals provide good data quality with low fitting errors and unique information on small scale variabilities in various BrO source regions such as Arctic sea ice, salt marshes, and volcanoes.

## 1 Introduction

Bromine monoxide (BrO) plays an important role in atmospheric chemistry. In the lower stratosphere, it is involved in chain reactions that deplete ozone (Wennberg et al., 1994) and bromine in the troposphere changes the oxidizing capacity through the destruction of ozone which is a primary precursor of atmospheric oxidation in the troposphere (von Glasow et al., 2004). In particular, large amounts of BrO are often observed in the polar boundary layer during springtime, known as "bromine explosion", and leading to severe tropospheric ozone depletion by autocatalytic reactions (McConnell et al., 1992; Simpson et al., 2007). In addition to polar sea ice regions, enhanced BrO concentrations were also detected over salt lakes/marshes (Hebestreit et al., 1999; Tas, 2005; Hörmann et al., 2016), in the marine boundary layer (Leser et al., 2003; Sander et al., 2003; Saiz-Lopez et al., 2004), and in volcanic plumes (Bobrowski et al., 2003; Theys et al., 2009; Schönhardt et al., 2017).

To understand the formation of BrO and the various chemical reactions involving halogen oxides in the troposphere, BrO observations have been carried out by in-situ chemical ionization mass spectrometry (CIMS) (Liao et al., 2011; Choi et al., 2012), ground-based differential optical absorption spectroscopy (DOAS) measurements such as long-path DOAS (LP-DOAS) (Hönninger et al., 2004; Liao et al., 2011; Stutz et al., 2011) and multi-axis DOAS (MAX-DOAS) (Hönninger et al., 2004; Frieß et al., 2011; Zhao et al., 2016). Using ground-based measurements, the diurnal variation and vertical distribution of BrO

can be investigated with high temporal resolution in specific source regions (Hönninger et al., 2004; Hendrick et al., 2007). However, ground-based measurements with localised spatial coverage are limited in observing large scale BrO explosion events and long-range transport of BrO plumes. This can be overcome by satellite measurements having extensive spatial coverage albeit at coarse spatial resolution and limited temporal sampling.

Since the launch in 1995 of the Global Ozone Monitoring Experiment (GOME) on ERS-2, a series of UV-visible spectrometers onboard satellites including SCIAMACHY, GOME-2, and OMI, have been used to monitor the global distribution of BrO columns and large-scale BrO events over time. The first global observations of BrO were retrieved from the measurements of GOME and large scale tropospheric BrO plumes in the polar sea ice region were detected (Wagner and Platt, 1998; Richter et al., 1998; Chance, 1998). SCIAMACHY which followed GOME not only measured BrO columns but also vertical profiles of

BrO in the stratosphere from limb measurements (Rozanov et al., 2005; Kuhl et al., 2008). The higher spatial resolution data of GOME-2 and OMI have been successfully used to monitor daily global distribution as well as BrO emissions from various source regions such as volcanoes (Theys et al., 2009; Hörmann et al., 2013; Schönhardt et al., 2017), salt lakes (Hörmann et al., 2016) and polar sea ice regions (Begoin et al., 2010; Salawitch et al., 2010; Theys et al., 2011; Sihler et al., 2012; Blechschmidt et al., 2016; Suleiman et al., 2018). However, OMI's coverage has been reduced since 2008 due to the so-called

"row anomaly", which is the result of a physical obstruction of the instrument, and currently, the anomaly effect extends over about 50% of the sensor's viewing positions (Torres et al., 2018). This reduced viewing ability affects the observation of emission events as well as the accuracy of the long-term time series. Existing satellite BrO time series can potentially be extended with data from the TROPOspheric Monitoring Instrument (TROPOMI) onboard the Copernicus Sentinel-5 Precursor platform, which was launched in October 2017 for a mission of seven years (Veefkind et al., 2012).

In this study, we present retrievals of BrO column amounts from TROPOMI observations on global and regional scales. This retrieval uses an optimized and adapted DOAS retrieval algorithm developed for earlier satellite missions. The aim of this study is a first demonstration of the feasibility of these new BrO retrievals on TROPOMI data, investigation of their precision and the comparison to data from other satellites. Therefore, the focus is on slant columns and simple vertical columns, determined using a geometric approximation for the air mass factor. Stratospheric correction schemes and more sophisticated

air mass factor calculations accounting for factors such as presence of clouds, varying surface albedo, and surface altitude are not used in this study. In order to determine the best retrieval window, sensitivity tests were performed for various measurement scenarios by a systematic investigation of retrieval results in different retrieval wavelength intervals in Section 3. TROPOMI BrO columns were assessed by comparison with those from the two existing satellite instruments, GOME-2B and OMI, with the consistency of the set of measurements being investigated. In addition, some examples of interesting cases in the

TROPOMI BrO data are identified and described for different source regions, such as the Arctic sea ice, salt lakes, and volcanoes.

## 2 S-5P/TROPOMI instrument

TROPOMI is a push-broom imaging spectrometer which was launched onboard of the European Space Agency's (ESA) Sentinel-5 Precursor (S-5P) satellite in October 2017 (Veefkind et al., 2012). The instrument has a large swath of 2600 km providing daily global coverage with high spatial resolution of currently 3.5x7 km$^2$ at nadir. TROPOMI has spectral bands in the ultraviolet (UV), visible (VIS), near-infrared (NIR) and shortwave infrared (SWIR), which allows it to monitor key atmospheric constituents such as $O_3$, $NO_2$, $SO_2$, CO, $CH_4$, HCHO, aerosols, clouds and various other trace gases. Compared to previous satellites, TROPOMI has prominent advantages in extended spectral band range and higher spatial resolution. From the 8 spectral bands of TROPOMI, band 3 data covering a spectral range of 320-405 nm with the spectral resolution of 0.5 nm and sampling of 0.20 nm/pixel (McMullan and van der Meulen, 2013) have been used for this BrO retrieval.

## 3 BrO retrieval

The retrieval algorithm for BrO uses the Differential Optical Absorption Spectroscopy (DOAS) technique (Platt and Stutz, 2008) as applied for space application (Burrows et al., 2011). The concept of DOAS is to separate the wavelength dependent extinction signal into two components, the low frequency and the high frequency part. The absorption by atmospheric gases are identified from their higher frequency structures of absorption cross-sections in spectral space and the low frequency parts are treated as a closure term fitted by a low order polynomial. The absorber concentration integrated along the light path, the slant column density (SCD), is determined assuming the Beer-Lambert law is applicable.

BrO SCD retrievals are typically performed within the wavelength range from 320 to 364 nm which covers 9 absorption peaks of BrO. In this spectral region, interferences with $O_3$ (Serdyuchenko et al., 2014), $NO_2$ (Vandaele et al., 1998), HCHO (Meller and Moortgat, 2000), $SO_2$ (Bogumil et al., 2003), OClO (Kromminga et al., 2003) and $O_4$ (Thalman and Volkamer, 2013) can be found. Thus, not only the absorption cross section of BrO (Wilmouth et al., 1999; Fleischmann et al., 2004) but also those of these related molecules are included in the BrO retrieval. In addition to the absorption cross-sections of interfering species, a synthetic Ring spectrum calculated using the SCIATRAN model (Vountas et al., 1998) is included to account for the effect of rotational Raman scattering and a linear intensity offset used as an additional closure term. All absorption cross sections are convoluted with TROPOMI's row and wavelength dependent slit function.

## 3.1 Sensitivity test of retrieval fitting intervals

For the retrieval of a weak absorbers such as BrO, the selection of the optimal fitting wavelength window is one of the most important things in the DOAS retrieval process (Vogel et al., 2013; Alvarado et al., 2014). The optimal fitting window is a retrieval wavelength range that maximizes the differential absorption structures for the trace gas of interest while minimizing interferences of other gases. In general, larger fitting windows can improve the quality of DOAS retrievals by using more spectral points, but at the same time, they can increase the noise and bias resulting from interfering signals with other absorbers and wavelength dependent light path lengths. Smaller fitting windows allow the fit to better compensate errors caused by

interferences of other absorption gases, but also can lead to increased cross correlation between reference absorption cross-sections and higher noise. Thus, finding a compromise for a fitting window that avoids the disadvantages as well as making the best use of the advantages from the retrieval wavelength interval is important to yield the best quality DOAS retrieval result.

In this study, a sensitivity test of the wavelength interval on DOAS BrO retrievals was performed by evaluating the BrO SCDs and fitting RMS values in many different wavelength ranges. In addition, the scatter of the slant columns was also investigated over a clean background region. Vogel et al. (2013) conducted a detailed study of the influence of the wavelength interval on the quality of DOAS retrievals based on a novel method and visualization of the results as contour plots. They applied this technique to a theoretical study of BrO retrievals for stratospheric BrO and BrO in volcanic plumes by using synthetic spectra,

modelling zenith-sky DOAS measurements of stratospheric BrO and tropospheric measurements of volcanic plumes. In this way, effects of different wavelength intervals on DOAS retrievals and appropriate spectral ranges for different study cases could be easily identified from visualized maps. A similar systematic approach was taken in this study. However, one important objective of this study is the investigation of the TROPOMI BrO retrieval results for various measurement cases and the identification of the overall best fitting window for TROPOMI BrO measurements. Therefore, sensitivity tests were performed

for different BrO emission scenarios using real satellite data to identify a spectral region with the best BrO results and the least interference problems. The different measurement scenarios selected are enhanced BrO plumes in the Arctic sea ice region, BrO plumes over a salt marsh, BrO enhancements in volcanic plumes, and clear and cloudy scenes over the Pacific background region. The selected scenarios have very different BrO amounts, slant columns of interfering species, solar zenith angle, temperatures, and geographical conditions. The influence of variations in the parameters for these different cases is included,

thereby enabling an optimal fitting window to be identified for application in a global BrO retrieval. The selected regions and dates for the different scenarios are summarized in Table 1. The sensitivity tests were performed over a wide range of retrieval wavelength intervals, which have start limits of 320-338 nm and end limits of 342-364 nm with an interval step of 0.2 nm (Fig. 1). This retrieval wavelength interval can contain up to 9 absorption peaks, always including at least the strongest absorption peak of BrO at 338-342 nm.

Apart from the retrieval wavelength range, other DOAS fit parameters were kept constant to isolate the effect of the retrieval wavelength range on the resulting BrO SCDs. The reference absorption cross-sections used in this sensitivity test include not only BrO but also the interfering species as discussed in section 3. Also, a row dependent daily earthshine radiance spectrum taken as the average of measurements over the Pacific region was used as a reference background spectrum to minimize across-track variability. In this sensitivity test, a polynomial of order 4 was used and kept constant, which is problematic for small

fitting windows for which a lower order polynomial might have been more appropriate, but changing the polynomial degree within the sensitivity test would have introduced another level of uncertainty. The DOAS retrieval technique is based on linear least squares fitting of spectra by minimizing the fitting residuals (chi-square values).

**Table 1.** Geographical and time information for the different scenarios of the sensitivity tests in Section 3.1.

|  | Latitude [°] | Longitude [°] | Date | No. of pixel |
|---|---|---|---|---|
| Polar sea ice | 72.5 ± 2.5 | 200.0 ± 20.0 | Mar 17 2018 (orbit# 2206) | 31261 |
| Salt marsh | 24.0 ± 0.3 | 70.0 ± 0.5 | Mar 31 2018 (orbit# 2397) | 137 |
| Volcanic plume | -16.0 ± 1.0 | 169.0 ± 1.0 | Nov 17 2017 (orbit# 492) | 1748 |
| Clear ocean | -7.0 ± 1.0 | -140.0 ± 14.0 | Apr 9 2018 (orbit# 2533) | 14254 |
| Cloudy scene | -3.0 ± 1.0 | -142.0 ± 14.0 | Apr 9 2018 (orbit # 2533) | 14255 |

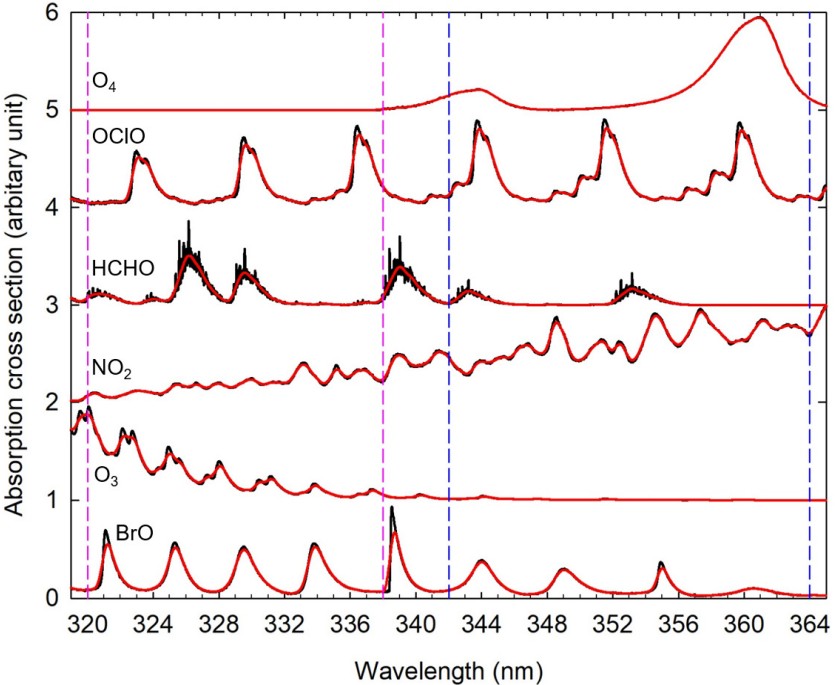

**Figure 1.** Reference absorption cross sections used in the sensitivity test of DOAS BrO retrieval. The spectra have been scaled to the order of 1 for presentation purposes. Black lines are the original cross sections and red lines are absorption cross sections convolved with the TROPOMI slit function (TROPOMI ISRF calibration key data v1.0.0). Pink vertical dashed lines indicate start wavelength ranges and blue lines end wavelength ranges of fitting windows for this sensitivity test.

### 3.1.1 BrO retrievals over the polar sea ice region

Satellite observations have shown large-scale BrO plumes (1000s of km) occurring over polar sea ice regions in spring, which indicates that this area is one of the most important BrO source regions (Simpson et al., 2007; Begoin et al., 2010). Thus, one of the retrieval wavelength interval sensitivity tests was performed for one BrO explosion event in the Arctic. The results are shown in Fig. 2, where each pixel corresponds to the mean of the retrieval results from a particular fitting wavelength interval

taken over the first region in Table 1 and is displayed on a colour coded scale. As can be seen in Fig. 2, negative BrO SCDs with relatively high fitting RMS values are found in general for BrO fitting windows with start wavelength below 327 nm and end wavelength below roughly 352 nm. These unphysical negative SCDs and high fitting RMS values may be attributed to interferences of other absorbers, which have strong absorption structures at shorter wavelength, in particular, $O_3$ which has a maximum at high latitudes in the spring season (Monks, 2000; Aliwell et al., 2002). This can potentially be improved by introducing additional ozone cross-sections, which attempt to account for effects arising from changes in the light path with wavelength (Pukite et al., 2010) (see Appendix A).

Also, the map of mean BrO SCDs shows a strong gradient near start wavelengths of 333.4 nm. As shown in Fig. 1, strong absorption features of $O_3$ are located at shorter wavelength range than 333.4 nm, which indicates that the sudden increase in BrO SCDs at the corresponding wavelength is likely due to the interference of gases other than $O_3$. In order to find the gas that interferes the most with the BrO retrieval, we investigated the retrieved SCDs maps of other reference gases used in this DOAS retrieval test and found that HCHO has a sharp change of SCDs in the vicinity of 333.4 nm similar to BrO with anti-correlations of both gases (Fig. 3). This implies that HCHO has a significant interference in the DOAS BrO retrieval at the wavelength range with a start limit of ~333.4 nm. Thus, to further examine the potential HCHO interference on the BrO retrieval, we performed additional sensitivity tests in the same way as before but excluding the HCHO absorption cross section for the Arctic BrO measurement scenario where very low HCHO columns are expected. The mean relative difference between BrO SCDs retrieved including the HCHO cross section (S1) and those retrieved without including HCHO (S2) is defined as:

$$\Delta_{rel}= 100~\% \times \frac{1}{N}\sum_{i=1}^{N}\frac{(S_{1i}-S_{2i})}{(S_{1i}+S_{2i})/2} \quad \text{(see Fig. 3)}.$$

Exclusion of the HCHO absorption cross section leads to reduction of the retrieved BrO SCDs at start limits above ~333 nm and in the range of start limits < 325nm, end limits < 351 nm, while an increase of BrO SCDs is observed mostly at the wavelength range with a start limit below ~333 nm. The pattern of variations in the retrieved BrO SCDs changes at the wavelength range between start limits of 333 and 333.4 nm where a strong absorption peak is present in BrO, while it is absent in HCHO. Possible artifact in the DOAS BrO retrieval caused by a spectral cross correlation between BrO and HCHO were also identified in Theys et al. (2011) and Vogel et al. (2013). From this sensitivity test for the polar BrO measurement scenario, we can confirm that main issues impacting the accuracy of the DOAS BrO retrievals are the influence of the strong $O_3$ absorptions at shorter wavelength range < 327 nm and potential interferences between BrO and HCHO absorptions. In consequence, we should choose a wavelength range that avoids strong $O_3$ absorption features as well as minimizing the interference between BrO and HCHO to obtain the most accurate BrO retrieval results.

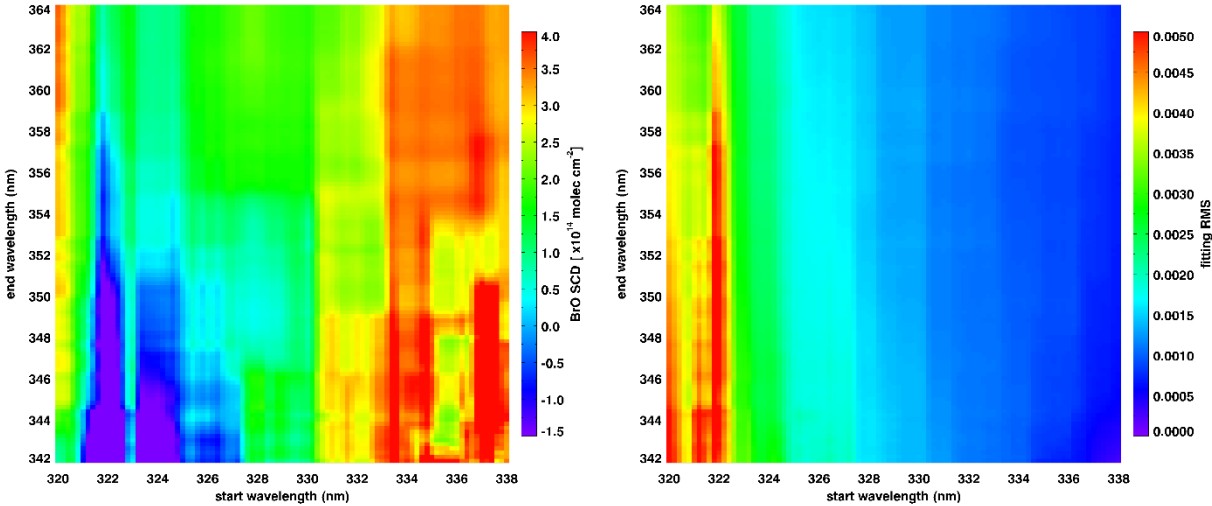

**Figure 2.** Colour coded means of BrO SCDs (left) and fitting RMS values (right) retrieved over the selected Arctic sea ice region for a BrO explosion event using TROPOMI measurements at different wavelength intervals.

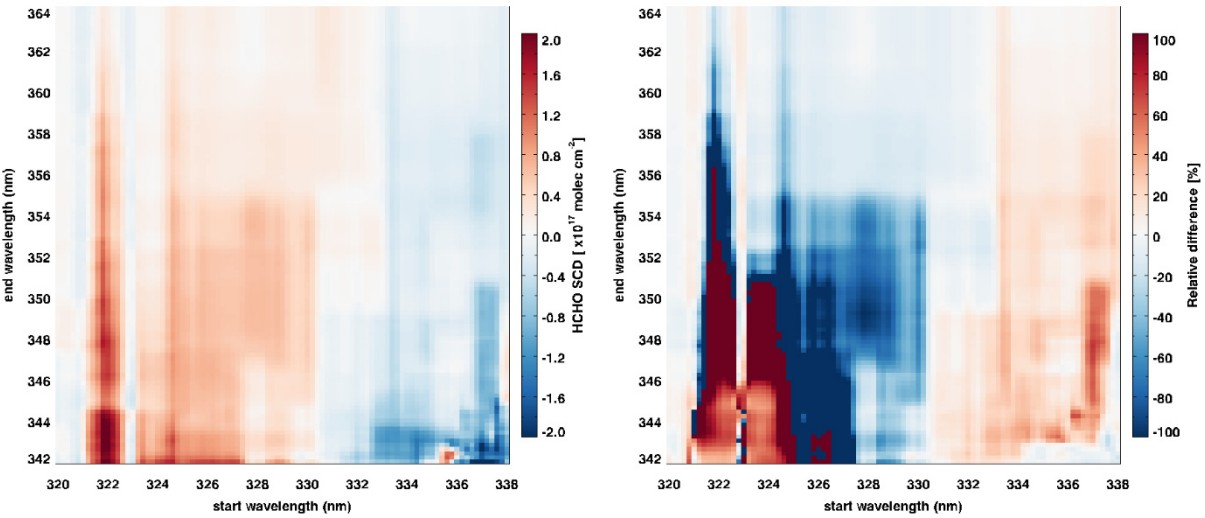

**Figure 3.** Colour coded means of HCHO SCDs (left) and mean relative difference between BrO SCDs retrieved including HCHO absorption cross section and those without HCHO (right) for the Arctic BrO measurement scenario.

### 3.1.2 BrO retrievals over a salt marsh

10   In addition to polar sea ice regions, salt lakes and marshes are important BrO source regions. The selected study region (see Table 1), Rann of Kutch, is known as one of the strongest natural sources of reactive bromine compounds and has been

monitored by satellite measurements for long-term variations of BrO columns (Hörmann et al., 2016). In order to determine the appropriate DOAS fitting wavelength range for BrO retrievals over salt marshes, the sensitivity test was performed in the same way as for the polar event. As shown in Fig. 4, BrO retrieval results in the salt marsh show relatively high fitting RMS values at shorter wavelengths below 322 nm, but unlike for BrO retrievals in the Arctic sea ice region (Fig. 2), no negative

values are found for BrO SCDs. This is because the interference of $O_3$ is smaller in this mid-latitude region scenario than in the polar region where the influence of $O_3$ absorptions is large. In general, high BrO SCDs with low fitting errors are shown in the evaluation wavelength range at start limits of 333-338 nm and end limits of 354-364 nm. This behaviour is similar to the appropriate retrieval wavelength range in the previous polar BrO sensitivity test results.

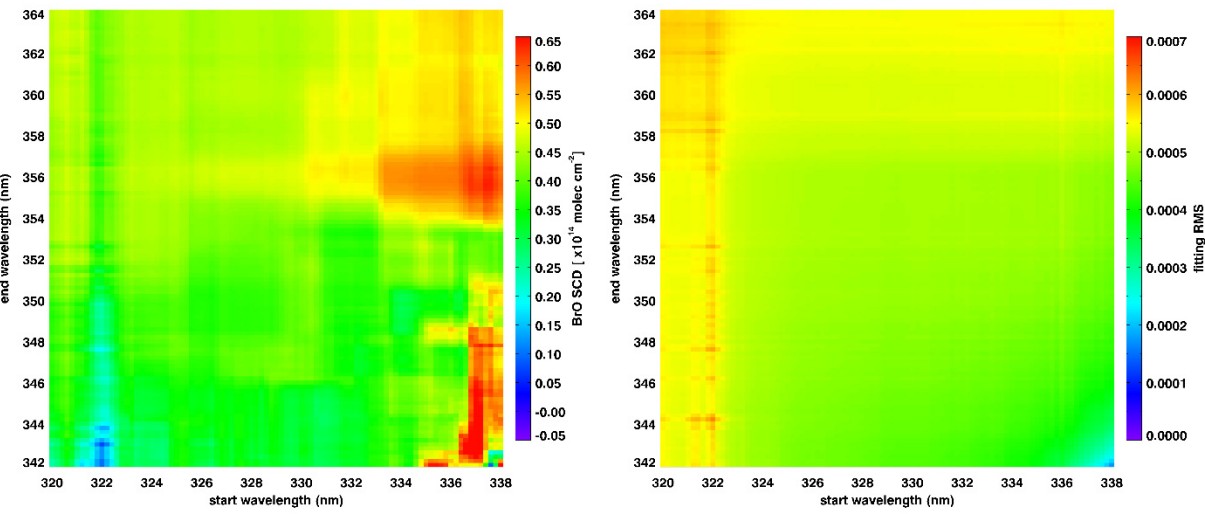

**Figure 4.** As Fig. 2 but for the Rann of Kutch salt marsh.

### 3.1.3 BrO retrievals in a volcanic plume

Volcanic eruptions emit various gases into the atmosphere and BrO is often also detected in volcanic plumes (Bobrowski et al., 2003). The selected volcanic BrO retrieval scenario is a small-scale BrO plume emitted by volcanic activity at Ambae. In

the sensitivity test for volcanic BrO, an $SO_2$ absorption cross section was added to the general DOAS BrO retrieval settings due to high $SO_2$ concentrations expected in the volcanic plume. If we use a retrieval wavelength interval with start wavelengths below 323 nm included or narrow fitting windows less than 8 nm wide, negative BrO SCDs and high fitting RMS values are found as can be seen in Fig. 5. These features may be attributed to the $SO_2$ interference at shorter wavelengths and the increase of cross correlation between BrO and other absorption gases, in particular, $SO_2$ (Fig. 6). Relatively higher fitting RMS values

are also found in the retrieval wavelength intervals extending to longer wavelengths ( $> 358$ nm). This is attributed to the impact of the Ring effect, i.e. the in-filling of Fraunhofer lines resulting from high aerosol loads and or the formation of clouds after the volcanic eruption (Theys et al., 2009) (see Fig.6).

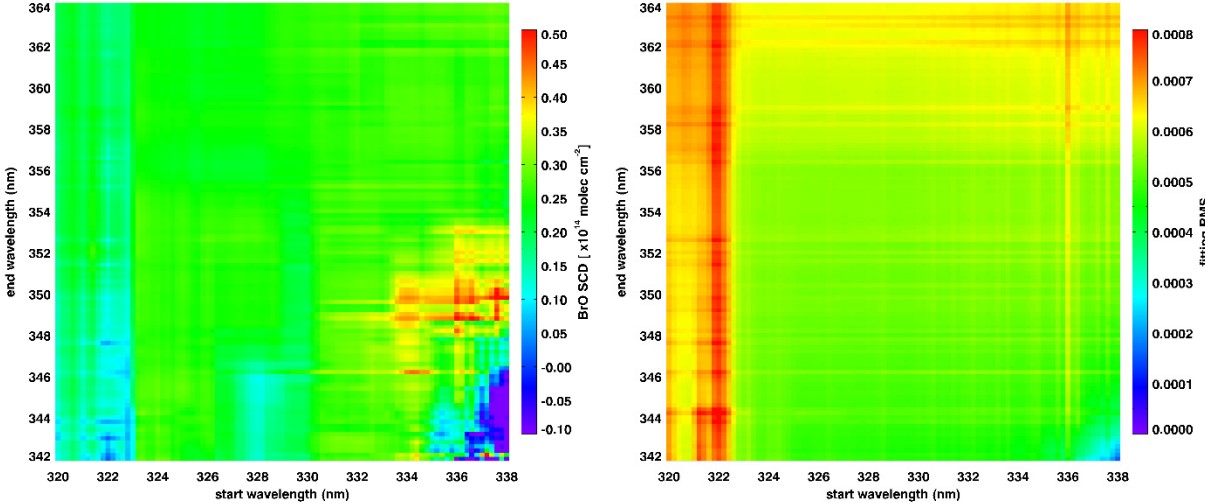

**Figure 5.** As Fig. 2, but for the selected volcanic plume case.

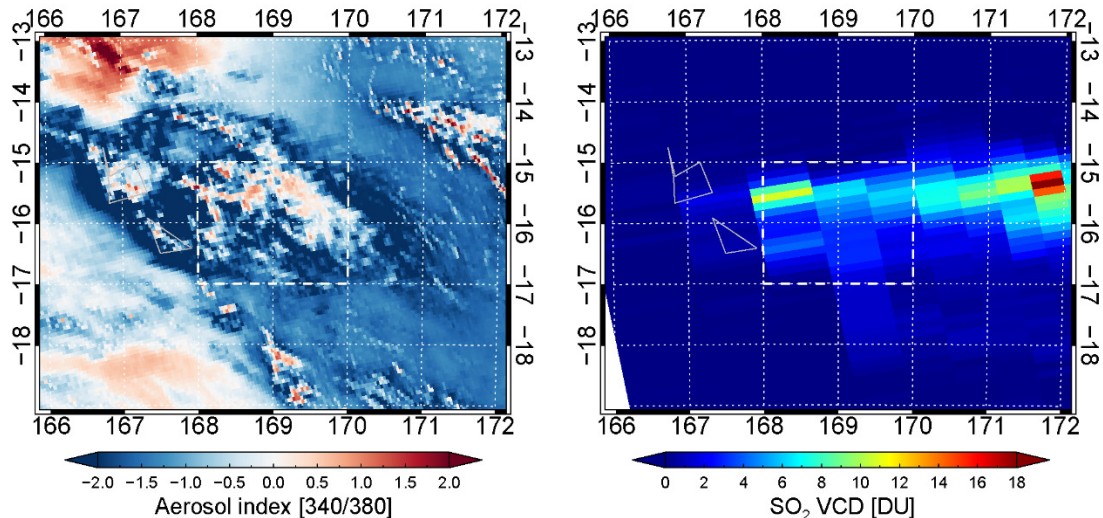

**Figure 6.** TROPOMI UV aerosol index (340 nm/380 nm) from the operational Level 2 product and OMI SO$_2$ vertical columns [DU] from the column amount SO$_2$ TRM (mid-troposphere) of the operational OMSO2 product for a volcanic BrO measurement scenario. The domain used for the sensitivity test is indicated by a gray dashed box.

### 3.1.4 BrO retrievals over clear scenes in the Pacific background region

The effect of different wavelength intervals on the BrO retrieval was also tested for the case of a clear scene in the Pacific background region without strong BrO sources and clouds. As this area is located within the background region used for the mean background spectrum, the BrO SCD should be minimal. As shown in Fig. 7a, in most of the retrieval wavelength intervals,

retrieved BrO SCDs are in fact close to the detection limit. Retrieval wavelength intervals having start wavelength smaller than 330 nm yield overestimations of SCDs, while retrieval wavelength intervals which start at wavelengths longer than 333.4 nm show mainly underestimations. In addition to the mean of the retrieved BrO SCDs, the root mean square error (RMSE) of the BrO SCDs in the clear background measurement scenario was also computed for each retrieval wavelength interval (Fig.

7b). The RMSE value represents the scatter of the BrO SCDs around the true BrO SCD, and thus a lower RMSE value indicates a better retrieval result with reduced uncertainty on the slant column. As can be seen in Fig. 7b, wider fitting windows show lower RMSEs, while more narrow fitting windows show higher RMSEs in general. This is reasonable, because if the fitting window is extended, we can exploit more spectral points in the retrieval and improve the quality of the columns from more available information (Richter et al., 2011). However, this advantage is reduced by the increasing importance of interfering

species, which is the reason for the increased RMSEs for fitting windows starting between 321 and 323 nm. In the BrO fitting RMS map, the values change abruptly at the wavelength of 322.6 nm, and high fit errors occur at wavelengths < 322.6 nm. This reduced fitting quality in the short wavelength range is attributed to the influence of absorption by stratospheric $O_3$.

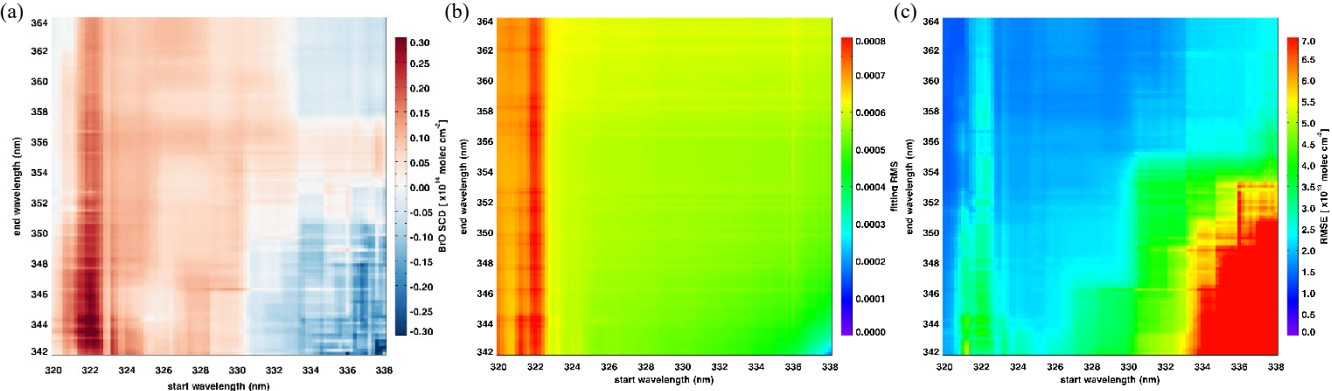

**Figure 7.** Mean values of (a) BrO SCDs, (b) fitting RMS values, and (c) root mean square deviation of BrO SCDs retrieved over the clear

part of the scene in the Pacific background region using TROPOMI measurements at different wavelength intervals.

### 3.1.5 BrO retrievals over cloudy scenes in the Pacific background region

In order to investigate the effects of cloud on the retrieval of BrO at different evaluation wavelength ranges, a cloudy area was selected in the Pacific background region and the sensitivity test was performed in the same way as in section 3.1.4. Figure 8

shows the means of retrieved BrO SCDs, root mean square errors of BrO SCDs, and fitting RMS values for the measurement scenario of cloudy scenes over the background region. The retrieved BrO SCDs in the cloudy scene are closer to the true value (0) than in the clear scene measurement scenario. In addition, the RMSEs for the cloudy sky measurement scenario is lower than for the clear sky case, but the variation pattern of RMSEs as function of the retrieval wavelength interval is similar. Both less over- or underestimations of the BrO SCDs and smaller deviations from the true BrO indicate lower uncertainties on the

cloudy sky BrO SCDs. This is expected because clouds are bright compared to the dark ocean surface and thus the instrument

receives a much larger signal. However, cloud effects are complex in DOAS retrievals using UV/vis measurements and the sensitivity depends on cloud properties such as cloud fraction, thickness, and top height (Burrows et al., 2011; Theys et al., 2011). The dependence of the retrieved BrO SCDs on the Ring effect due to the presence of clouds is also shown in the map of fitting RMS values (Fig. 8b). Unlike the fitting RMS variations in the cloud-free condition (Fig. 8b), the cloudy sky measurement scenarios show relatively higher fitting RMS values at wavelengths longer than 358 nm, in agreement with the findings for the volcanic plume.

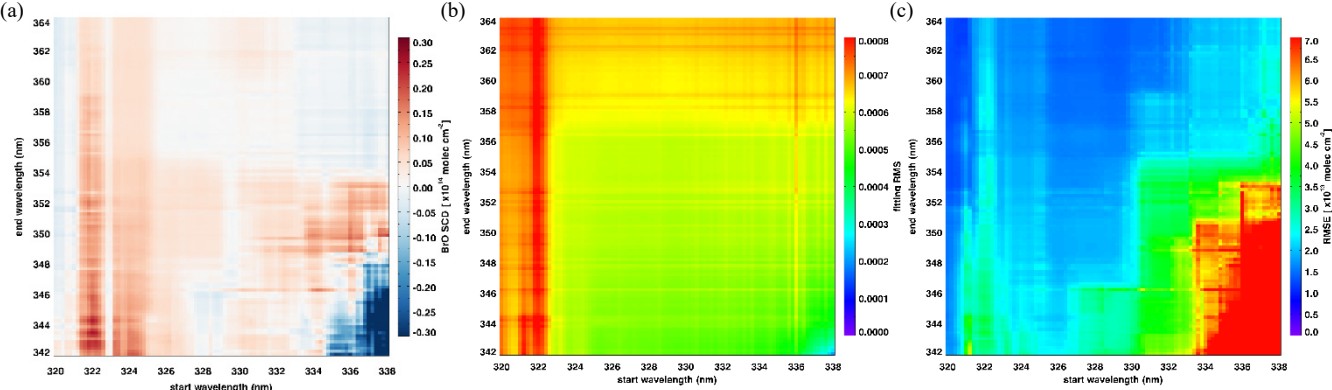

**Figure 8.** As Fig. 7 but for the cloudy part of the Pacific background region.

### 3.1.6 Selection of optimal fitting window

In the previous sections, the influence of the retrieval wavelength interval on the DOAS BrO retrieval was tested for different measurement scenarios. Based on these test results, we can determine the best fitting window for TROPOMI BrO retrievals for global analysis as well as for the primary BrO source regions. The optimal retrieval fitting windows can be defined as those wavelength intervals which show higher BrO signals with lower fitting residuals in the BrO source regions, while the BrO SCDs should be minimal with narrow distributions of SCDs over the clean Pacific background region. The test for polar BrO retrievals showed unphysical values in wavelength intervals including lower wavelengths smaller than 327 nm due to strong $O_3$ interferences. The effect of $SO_2$ interference with strong absorptions at lower wavelengths was also confirmed in the volcanic BrO measurement scenario. Their interfering influences are reduced at longer wavelengths as overall absorption cross section structures decrease, thus retrieval wavelength intervals with a start limit above 327 nm are preferred to avoid the strong dependency on the lower wavelength limit. In addition to $O_3$ and $SO_2$ absorptions at shorter wavelengths, HCHO can also interfere in DOAS BrO retrievals through anti-correlation between the two gases especially at ~333 nm. This potential artefact may be attributed to the cross correlation caused by the absorption band shape of BrO and HCHO, and it is necessary to find a retrieval wavelength interval that minimizes possible errors caused by the BrO-HCHO cross correlation. In the case of DOAS retrievals over the cloudy background region and the volcanic plume, higher fitting errors were found in the wavelength intervals extending beyond 362 nm because of imperfect correction of the Ring effect and possibly also poorer fitting of $O_4$

related to the temperature dependency of the cross section. While minimizing these sources of uncertainty on the retrieval, the range of reasonable BrO SCDs and low retrieval errors for all measurement scenarios are observed in the wavelength range of start limits of 334-338 nm and end limits of 358-362 nm. Finally, the fitting window 334.6-358 nm was selected for TROPOMI BrO retrievals with other retrieval parameters set as shown in Table 2, by comparing fit residuals and SCD distributions for

5   the remaining fitting windows. Figure 9 shows a spectral fitting example of a pixel from orbit 2206 on 17 March 2018 passing over Arctic sea ice region (72.55 °N, 200.40 °E in Fig. 12c). The larger BrO SCD of $4.72 \times 10^{14}$ molec cm$^{-2}$ was retrieved with relative small fitting error of 6.2 %. Although the choice of the optimal fitting window may seem arbitrary to some degree, the analysis of several different relevant scenarios for many possible combinations of fitting windows described above demonstrates that this is an overall robust selection. However, further studies are needed to address the remaining challenges

10  identified through the sensitivity tests, in particular the possible spectral cross correlation of BrO with HCHO around the selected fitting window.

**Table 2.** DOAS settings used for the BrO slant column retrievals and instrumental inter-comparison.

| Parameter | Description |
| --- | --- |
| Fitting window | 334.6-358 nm |
| Absorption cross-sections | BrO (Wilmouth et al., 1999), 228 K<br>$O_3$ (Serdyuchenko et al., 2013), 223 and 243K<br>$NO_2$ (Vandaele et al., 1998), 220 K<br>OClO (Kromminga et al., 2003), 213 K<br>HCHO (Meller and Moortgat, 2000), 298 K<br>$O_4$ (Thalman and Volkamer, 2013), 293 K |
| Ring effect | Ring cross section calculated by SCIATRAN model |
| Polynomial | 5 coeff |
| Solar reference spectrum | Kurucz solar spectrum (Chance et al., 2010) |
| Background spectrum | For TROPOMI and OMI one spectrum per row, daily averaged earthshine spectrum in selected Pacific region |
| Intensity offset correction | Linear offset |

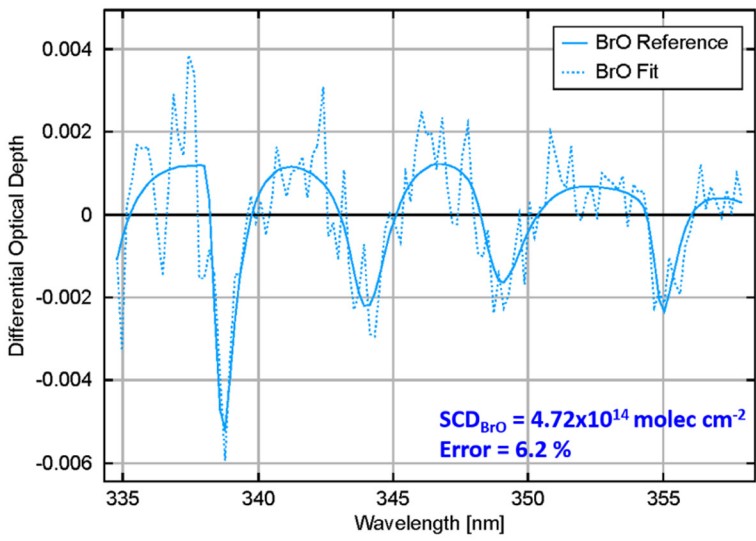

**Figure 9.** Example of a BrO fit result applying DOAS settings of Table 2 in the Arctic BrO measurement case. The dashed line shows fit results including the fitting residual and the solid line is the reference spectra scaled according to the fit result.

### 3.2 Destriping and offset correction

TROPOMI is an imaging spectrometer operating in push-broom configuration where one direction of the two-dimensional CCD detector is used for the wavelength axis and the other for the across-track image of the 2600 km wide instrument's field of view. This is similar in concept to OMI but with 450 instead of 60 spatial rows at much higher spatial resolution (Veefkind et al., 2012). In this instrument configuration, across-track variability can appear as stripes in trace gas columns due to small variations between the rows which are not completely compensated by lv1 calibration. Indeed, OMI has shown this across-

track striping problem (Boersma et al., 2007) and explicit destriping is applied in many OMI products. In TROPOMI data, stripes are also apparent in some trace gas maps when using solar irradiance measurements as background. Two different approaches can be carried out to correct for this: Either, the across track variability is determined on daily basis over a region with minimal variability in trace gas columns and subtracted from all retrieved slant columns, or irradiance background spectra are replaced by averages of nadir observations taken over a reference region. In this study, the second approach is used and

daily row-dependent mean radiances measured over a selected Pacific region (30°S-30°N, 150-240°E) are used as background spectrum. This approach can effectively remove across-track stripes, but the retrieved differential BrO SCDs have to be corrected for the viewing angle dependency of the column over the reference region. Here, BrO SCDs were normalized to an assumed background level of a BrO VCD of $3.5 \times 10^{13}$ molec cm$^{-2}$ over the Pacific background as suggested by previous studies (Richter et al., 2002; Sihler et al., 2012) using a two-step approach (see Fig.10). First, an offset value for normalization of the

differential BrO SCDs is determined as the mode of the Gaussian distribution of differences between the differential SCDs in the reference sector and the normalized SCDs estimated by multiplying the background VCD and a geometric air mass factor

defined as $\text{AMF}_{\text{geo}} = \frac{1}{\cos(\text{SZA})} + \frac{1}{\cos(\text{VZA})}$. In a second step, this offset value is modified for each row depending on the viewing zenith angle (VZA) to account for variations of the BrO air mass factor. The normalized SCDs are finally calculated by subtracting the VZA-dependent offset values from the measured SCDs.

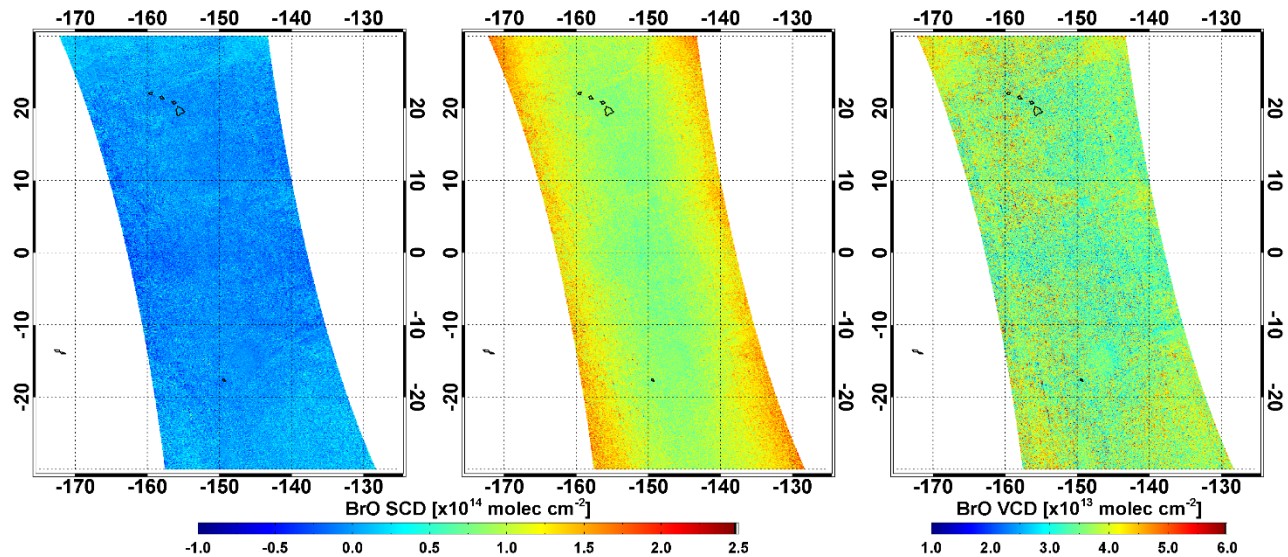

**Figure 10.** Illustration showing destriping and offset correction steps described in section 3.2 using TROPOMI orbit 2207 on Mar 17 2018. BrO SCDs retrieved by daily row-dependent mean radiances in the Pacific reference sector as background spectrum for the across-track correction (left), offset-corrected BrO SCDs treated by applying the normalization approach including the VZA dependency on the BrO SCDs (middle), BrO VCDs computed by dividing the offset-corrected BrO SCDs by geometric AMFs (right).

## 4 Results

### 4.1 Global distribution of BrO

Applying the retrieval settings described in Table 2, BrO vertical columns have been computed from TROPOMI, OMI and GOME-2B Level 1 spectra. It should be noted that BrO VCDs were calculated using geometrical air mass factors and data for
15   solar zenith angles larger than 85° and chi-square values greater than 0.01 were excluded. For OMI, ground/atmospheric scenes affected by the row anomaly were also excluded by using the OMI XTrackQualityFlags. The data in row #42 to 45 were additionally ignored because it was apparent from the BrO SCDs that they were affected by the row anomaly although they were not marked as bad pixels. No cloud screening was applied. Figure 11 shows the global distributions of the monthly averaged BrO total vertical columns from the three satellites for April 2018. The spatial distributions of BrO columns show a
20   good consistency in spite of the differences in instrument resolution and overpass times. High BrO values are shown in the northern high latitude region because of tropospheric bromine explosions over the Arctic sea ice during spring time as discussed in previous studies (Richter et al., 2002; Simpson et al., 2007; Begoin et al., 2010), whereas BrO values are low in

the tropics and mid-latitudes where BrO columns are primarily of stratospheric origin. Relatively higher BrO values are found in the subpolar and Antarctic region compared to tropics and mid-latitudes. This might reflect real BrO column increases but could at least partially be related to the use of geometric AMFs which do not consider the effects of surface albedo and clouds. The number of photons detected at the satellite is larger over bright surface areas than over dark surface areas. Therefore, the
use of a simple AMF which does not consider the sensitivity to surface albedo can underestimate BrO vertical columns over dark surfaces such as the ocean in comparison to high surface albedo regions such as the Antarctic region, north of Russia and Canada. In addition to the surface albedo, clouds also affect signals detected at the satellite. The light path length and intensity are significantly changed by the cloud top height, cloud thickness and cloud fraction. Using an AMF that does not take into account the cloud effects can therefore result in errors in the computed vertical columns, as can be seen from the slightly higher
BrO VCDs in the subpolar regions where cyclones are frequently observed due to the sub-polar low pressure system (Fig.11). Consequently, an improved AMF reflecting the sensitivity of surface albedo, cloud properties and BrO vertical profile should be calculated to obtain more accurate vertical column densities (Theys et al., 2011; Sihler et al., 2012), and this will be investigated in detail in a follow-up study using surface albedo and cloud information from the operational satellite products as they become available.

To assess the random noise of the BrO retrievals for the different instruments, distributions of SCDs and retrieval fitting RMS values over a clean Pacific region (10°S-10°N, 150-260°E) were analysed for April 2018. Here, differential BrO SCDs without the background offset correction were used for more clear interpretation. As shown in Fig. 11b, all three satellite BrO SCD distributions show nearly Gaussian shape and are centered around zero with FWHMs of 0.50, 0.80, and $0.79 \times 10^{14}$ molec cm$^{-2}$ for TROPOMI, GOME-2B, and OMI, respectively. However, while TROPOMI and GOME-2B columns are symmetrically
distributed close to the detection limit, OMI data are slightly shifted towards positive values. The latter is attributed to be a consequence of systematic biases caused by the relatively lower quality of Level 1b radiance due to the instrument degradation. TROPOMI shows the smallest scatter of BrO SCDs, with OMI and GOME-2B having about 60% larger FWHMs. TROPOMI retrievals also show by far the smallest mode of the fitting RMS distributions, demonstrating the excellent signal to noise ratio per pixel even at the unprecedented small foot print.

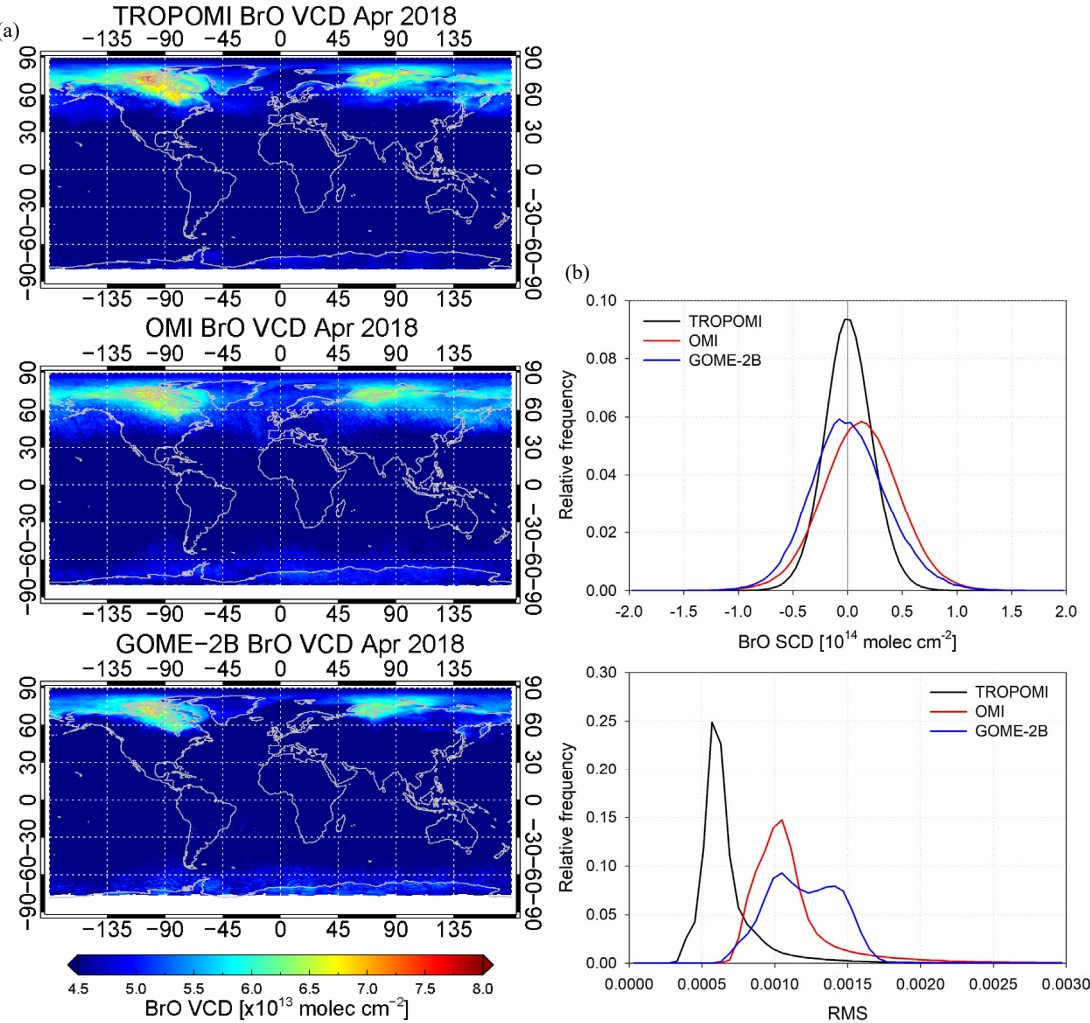

**Figure 11.** (a) Global distributions of monthly mean BrO vertical columns retrieved from TROPOMI, OMI and GOME-2B measurements for April 2018. Data with a solar zenith angle lower than 85° were used and in the case of OMI data, only data not affected by the row anomaly were included. (b) Distribution of BrO SCDs and fitting RMS values over a clean equatorial Pacific region (10°S - 10°N, 150 - 260°E) for the same study period.

## 4.2 Comparison to OMI and GOME-2B retrievals

To evaluate the consistency of TROPOMI BrO retrievals with those from other satellites, a comparison of BrO satellite retrievals was performed using GOME-2B and OMI retrievals obtained by applying the same retrieval setting (Table 2) to Level 1b data. However, for the comparison of different satellite retrievals, several things have to be considered. First of all, the three satellites have different spatial resolution, $40 \times 80$ km$^2$ for GOME-2B, up to $13 \times 24$ km$^2$ for OMI, and $3.5 \times 7$ km$^2$ for TROPOMI. To establish a relationship between different satellite values with different pixel sizes, a spatial coupling of the different datasets is required. Here, the higher spatial resolution TROPOMI data was averaged based on a grid of lower spatial resolution. Each GOME-2 and OMI BrO measurement was compared to the averaged TROPOMI BrO that lay within the

distance of 0.3 and 0.1 degree from their center of pixels, respectively. In addition to different pixel sizes, the effect of different overpass times between satellites should be considered. TROPOMI which has an ascending orbit with a local equator crossing time of 13:30 LT shows different overpassing time than GOME-2B which has a descending node equatorial crossing time at 9:30 LT, whereas it has a similar afternoon overpassing time with OMI. Although having a similar overpassing time on the

5    ascending node to TROPOMI, recent OMI data provide only limited data due to the loss of spatial coverage with the expansion of the row anomaly, especially in the middle and east across-track segment of the orbit (Torres et al., 2018). This led to a difficulty in utilizing orbits having similar measurement time for the two satellite instruments.

Fig. 12a shows a scatter plot comparison between TROPOMI and OMI BrO VCDs, and Fig. 12b compares TROPOMI and GOME-2B BrO VCDs. As mentioned before, BrO VCDs were converted from SCDs by dividing geometric AMFs. The

10   comparison was performed for enhanced BrO plumes in the Arctic sea ice region on March 17th 2018 (Fig. 12c). Despite the different spatial resolutions and measurement times of the instruments, TROPOMI BrO shows good agreements with both OMI and GOME-2B BrO with correlations of 0.84 and 0.84, and slopes of 0.89 and 0.72, respectively. This good agreement and consistency of TROPOMI data with previous satellite sensors suggest that these data could be used to extend the existing long-term data set of space-based BrO observations, in particular for tropospheric BrO explosion events.

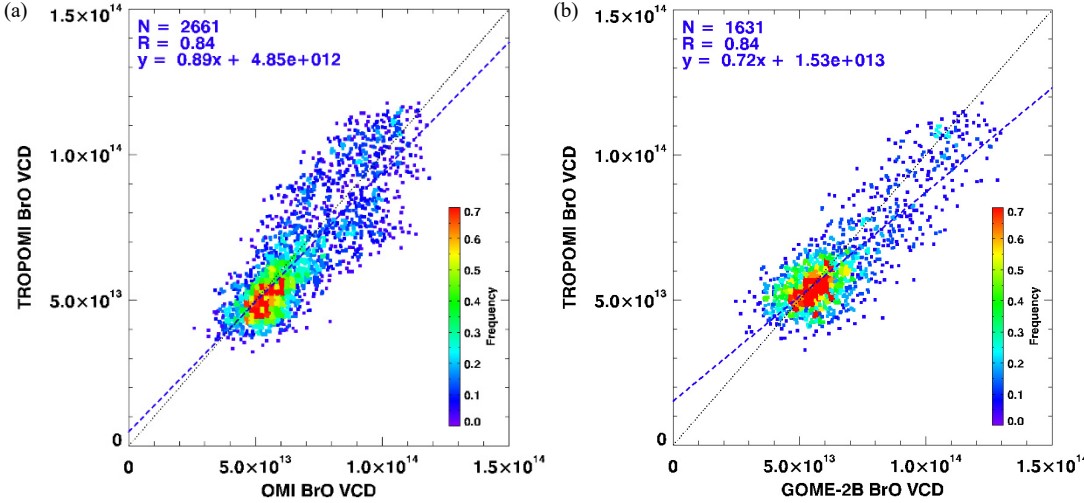

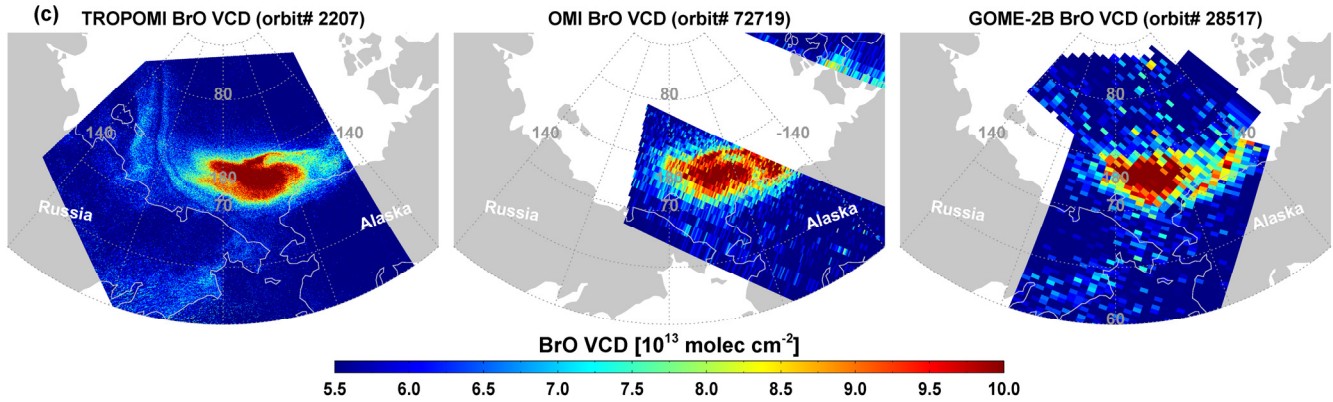

**Figure 12.** Scatter plots of (a) TROPOMI and OMI BrO vertical columns, (b) TROPOMI and GOME-2B BrO vertical columns in (c) the selected region of enhanced BrO plumes on March 17, 2018.

### 4.3 BrO observations in tropospheric source regions

#### 5 4.3.1 BrO plumes over Arctic sea ice

Explosive enhancements of BrO in the troposphere taking place in the polar boundary layer during spring have been reported from ground-based measurements and satellite observations (Hönninger et al., 2004; Begoin et al., 2010; Choi et al., 2012). As an illustration of the signature of such events in TROPOMI data, Figure 13 presents maps of the TROPOMI, OMI and GOME-2 measurements of total column BrO on 10 April 2018. A small compact BrO enhancement as well as a long BrO

plume extending along the coastline can be identified in the figures. The long and thin enhanced BrO plume near the coastline is prominent in the map of TROPOMI, while it can hardly be discerned in the OMI and GOME-2B maps. For the OMI retrievals, a significant part of the scene is missing because of filtering for pixels affected by the row anomaly. The GOME-2B orbit shown was taken about 1 hour before the TROPOMI and OMI measurement times but the BrO plumes are detected in similar locations and have comparable size as in the TROPOMI data. However, the details of the spatial distribution and

plume shape cannot be confirmed due to the lower spatial resolution of GOME-2.

Another example of a BrO explosion event case is shown in Figure 14. A relatively narrow and long shape of enhanced BrO over the Beaufort Sea can be found in all three satellite maps. As discussed for the previous example, TROPOMI data with the high spatial resolution of $3.5 \times 7$ km$^2$ yield a more detailed view of the BrO explosion event compared to OMI and GOME-2B. The enhanced BrO plumes appear around open leads and sea ice cracks shown as slightly darker areas in the matching

MODIS image (arrows pointing at examples). In particular, the elevated BrO around the Banks Island and the eastern Beaufort Sea (-140~-120 °E, 70~77 °N) could be significantly linked to open leads since frost flowers and sea salt aerosols which act as the source of reactive bromine can be formed in such areas (Simpson et al., 2007). Also, opening of sea ice leads can locally create enhanced vertical mixing and uplifting of bromine sources. However, the analysis of the long enhanced BrO plume from the coast of Alaska towards north should be cautious. The MODIS image composed of the 7-2-1 bands can distinguish clouds

(as white) from the sea ice (as sky blue), and this image shows that the shape of the enhanced BrO plume is similar to that of

clouds. Convective clouds can be formed around open leads due to the supply of water vapor and enhanced vertical mixing, but computed BrO enhancement over clouds may have an error because of the use of AMFs which do not consider the effects of clouds. In spite of this uncertainty, the enhancement of vertical columns by up to $4\times10^{13}$ molec cm$^{-2}$ compared to the surrounding values indicates that open leads could be associated to the BrO enhancement. Small-scale BrO explosion events around open leads or polynyas can be better investigated with the high spatial resolution TROPOMI data and will be the topic of a follow-up study.

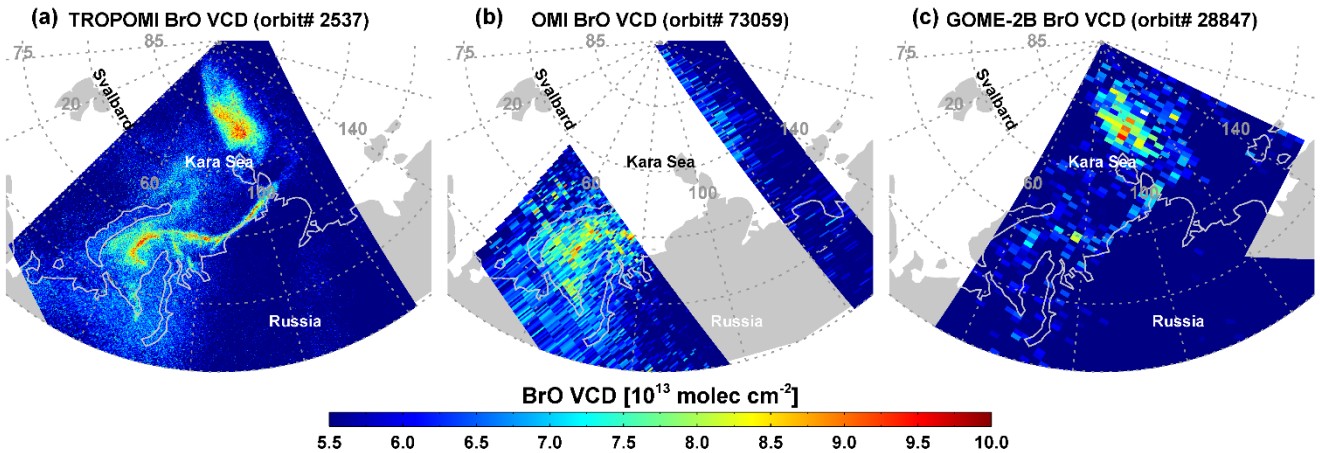

**Figure 13.** BrO geometric vertical columns observed over the Arctic sea ice region on 10 April 2018 by (a) TROPOMI, (b) OMI and (c) GOME-2B.

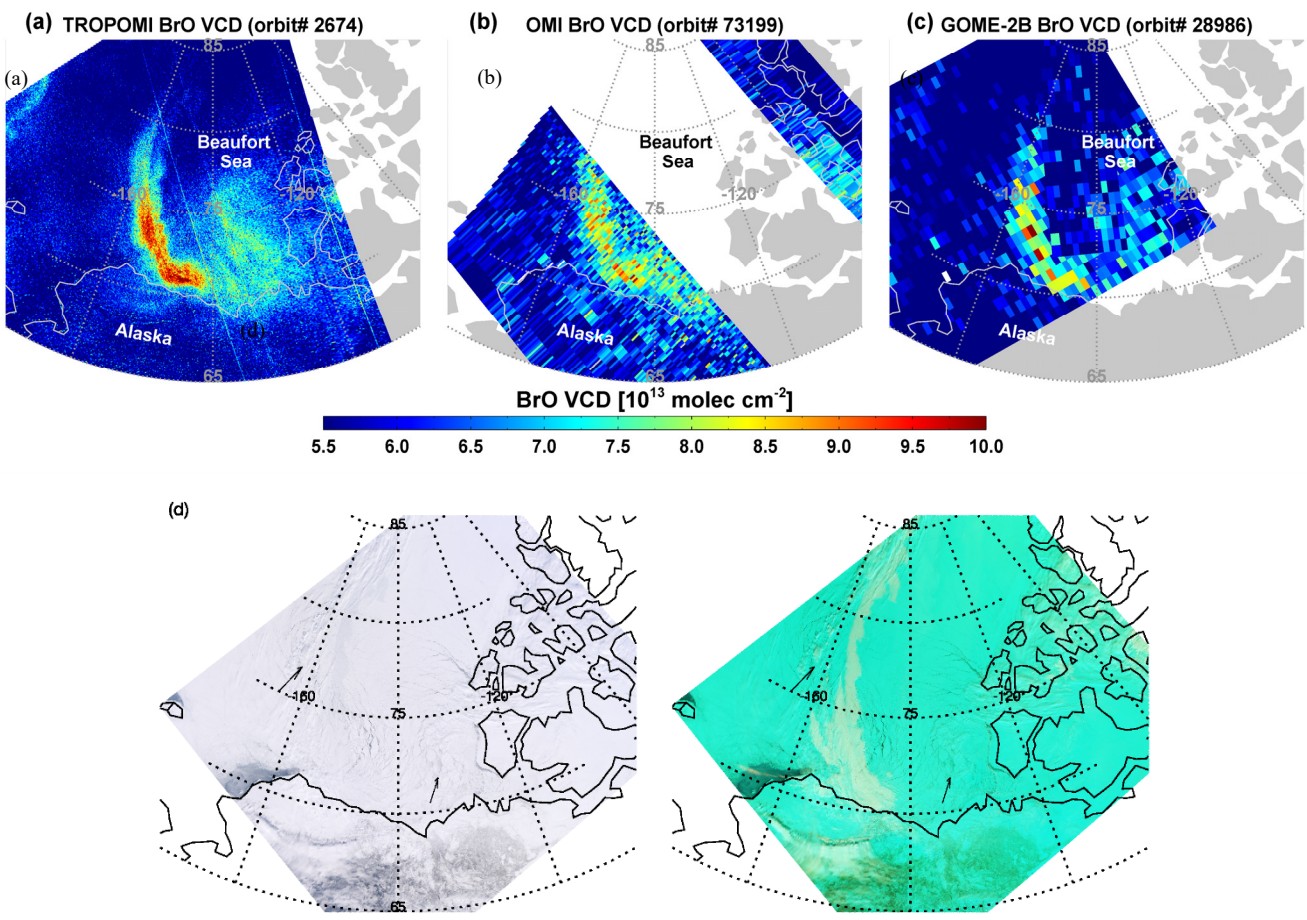

**Figure 14.** BrO geometric vertical columns observed over the Arctic sea ice region on 19 April 2018 from (a) TROPOMI, (b) OMI and (c) GOME-2B, respectively. (d) MODIS true colour image (left) and image using combinations of 7-2-1 bands (right) from the Aqua satellite for the same scene on the same day. Leads are slightly darker in the MODIS image as indicated by the arrows.

### 4.3.2 BrO plumes over a salt lake/salt marsh

Salt lakes are one of the strongest and most localized natural sources of reactive bromine. Consequently, BrO amounts over multiple salt lakes and marshes have been investigated by ground-based DOAS instruments and satellites. High BrO concentrations with peak mixing ratios of 86 ppt at the Dead Sea were observed by Long-Path DOAS measurements in 1997 (Hebestreit et al., 1999), followed by studies of the diurnal cycle of BrO and the relationships between BrO and $O_3$ and meteorological factors in the Dead Sea region (Matveev et al., 2001; Tas, 2005). BrO over salt lakes was also studied using satellite measurements. Chance (2006) showed BrO enhancement over the Great Salt Lake using OMI measurements and Hörmann et al. (2016) found a typical annual BrO formation cycle over the Rann of Kutch seasonal salt marsh using long-term GOME-2 and OMI data.

The release of reactive bromine and enhanced BrO plumes over the Rann of Kutch salt marsh are also readily detected by TROPOMI data. Daily mean BrO VCDs over the Rann of Kutch area for TROPOMI and OMI, and MODIS true color images for the time period from 11 April to 14 April 2018 are shown in Figure 15. It should be noted that the AMFs used in this work do not consider surface albedo and cloud effects, and therefore BrO VCDs may be overestimated over the bright salt marsh.

5  BrO enhancements of up to $4.5 \times 10^{13}$ molecules cm$^{-2}$ over background values are detected as hotspots in both satellites. However, as can be seen in Fig. 15, TROPOMI data show BrO plumes and small scale variabilities much more clearly with more spatial details than OMI data. In case of OMI data, BrO plumes are detected by only a few pixels, whereas TROPOMI can detect the same plumes by hundreds of data points (~150 pixels). This illustrates that TROPOMI data will facilitate in depth studies of localized small-scale BrO events for multiple salt lakes and marshes.

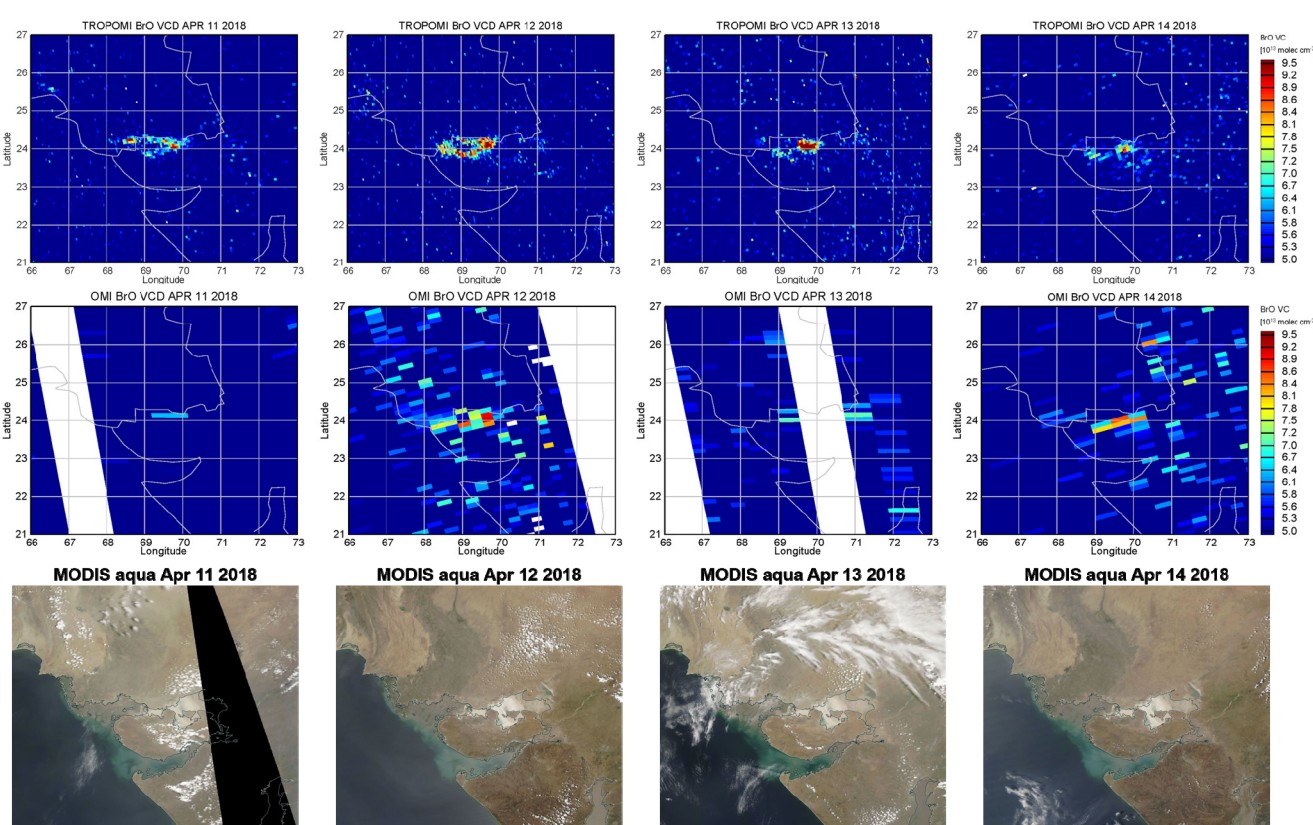

**Figure 15.** Daily BrO geometric vertical columns [$10^{13}$ molec cm$^{-2}$] over the Rann of Kutch salt marsh on 11, 12, 13, and 14 April 2018 from TROPOMI (top) and OMI (middle) measurements as well as MODIS Aqua true colour images over the study region for the same days (bottom).

15  **4.3.3 BrO enhancements in volcanic plumes**

Explosive volcanic eruptions lead to the formation of BrO into the troposphere and lower stratosphere. The detections of BrO in volcanic plumes has been reported by ground-based DOAS measurements for several volcanoes (Bobrowski et al., 2003;

Oppenheimer et al., 2006; Boichu et al., 2011). In addition to ground-based measurements, BrO in a volcanic plume was first detected in GOME-2 satellite data after the eruption of the Kasatochi volcano by Theys et al. (2009). Following the first satellite volcanic BrO detection, Hörmann et al. (2013) investigated 64 volcanic plumes and BrO/SO$_2$ ratios using GOME-2. Schönhardt et al. (2017) found not only volcanic BrO but also IO emissions using SCIAMACHY and GOME-2 measurements.

Not surprisingly, volcanic plumes containing BrO are also detected in TROPOMI data. Figure 16 shows the plume over the Indonesian island of Bali after a volcanic activity at Mount Agung on 29 November 2017. Enhanced BrO values of up to 8.5x10$^{13}$ molec cm$^{-2}$ and dispersion of plumes by the wind towards the Southwest were detected by both TROPOMI and OMI. These volcanic BrO plumes are associated with enhanced SO$_2$ values as identified from the NASA operational OMI product (ColumnAmountSO2_TRM in OMSO2 version 3 product) with a positive correlation between the species. As shown in Fig.

16, monitoring of BrO emissions and their relationship to other gases from volcanic activities is possible with TROPOMI data at higher spatial resolution and improved sensitivity, which suggests that more detailed analysis of volcanic BrO will be possible in the future.

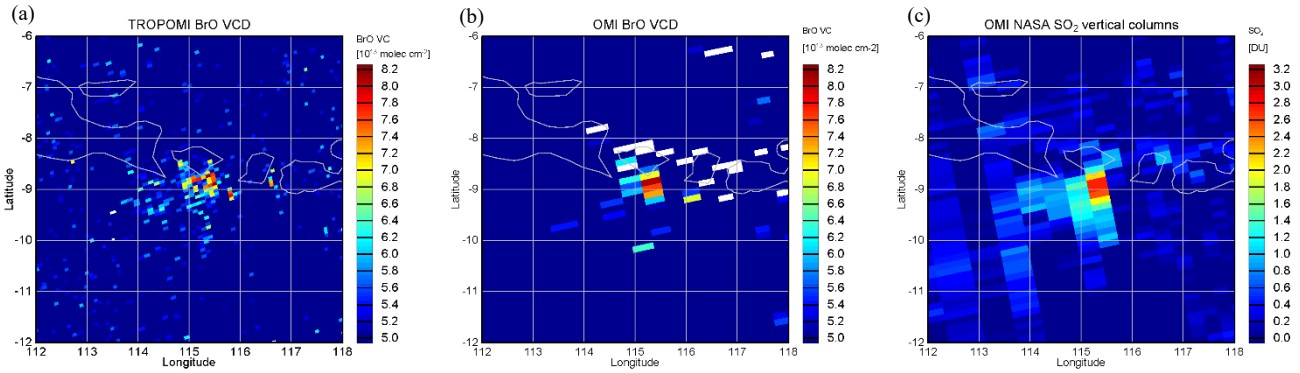

**Figure 16.** Volcanic BrO vertical columns [10$^{13}$ molec cm$^{-2}$] on 29 November, 2017 after volcanic eruptions at Mount Agung on the Indonesian island of Bali observed by (a) TROPOMI and (b) OMI. (c) Volcanic SO2 vertical columns [DU] from the column amount SO2 TRM (mid-troposphere) of NASA operational OMSO2 product (https://disc.gsfc.nasa.gov/datasets/OMSO2_V003/summary).

## 5 Conclusions

Adapting and optimizing the DOAS retrieval developed for earlier satellite missions, a first BrO column retrieval for measurements of TROPOMI, the new space borne instrument launched on the European Sentinel 5 Precursor satellite in October 2017, was developed. One of the most important factors in the DOAS retrieval is the wavelength interval selected as the fitting window with the objective being to maximize the differential absorption structures of the specific gas of interest and minimize the influence of other interfering signals. However, finding the optimal retrieval wavelength interval is not

straightforward as instrumental factors as well as viewing conditions and study area can impact on the results. Similar to the approach by Vogel et al. (2013), colour coded maps of DOAS retrieval results obtained by systematically varying the retrieval

wavelength interval were created for various observation scenarios on TROPOMI data to determine the optimal retrieval wavelength interval for BrO. Negative BrO SCDs, large deviations from the expected BrO SCDs and high fitting errors occur at shorter wavelengths when strong absorption structures of $O_3$ and $SO_2$ are included. The sensitivity of BrO retrieval to HCHO was also found by showing anti-correlation between two gases. At wavelengths longer than 362 nm, poorer results were found

in the cloudy scene and volcanic plume measurement scenario, presumably due to the wavelength dependency of the Ring effect and imperfect fitting of $O_4$. Based on the information gathered from the sensitivity tests for different measurement scenarios, 334.6-358 nm was selected as the optimal wavelength range for TROPOMI BrO retrievals for most of the possible measurement situations. This fitting window yields reliable BrO retrieval results with small fitting errors, but future studies on quantitative assessments and cross correlations between BrO and all the interfering absorbers are encouraged to further

improve retrieval results. As imaging instruments such as TROPOMI often show across track offsets in the retrieved columns, the DOAS BrO retrieval has to include a destriping method. In this study, row specific daily averaged earthshine radiances from a Pacific background area are used as reference spectrum in the DOAS fitting procedure and a post-processing offset correction is applied to convert the resulting differential slant columns to absolute slant columns. Conversion to vertical columns is achieved in this study by using a simple geometric air mass factor.

As a first consistency test, TROPOMI BrO columns were compared with OMI and GOME-2 data on both global and regional scale. TROPOMI BrO retrievals show good agreements with OMI and GOME-2B BrO columns with high correlation coefficients (slopes of the regression lines) of 0.84 (0.89) and 0.84 (0.72) for enhanced BrO plumes in Arctic sea ice region. Global maps of monthly BrO columns also agree well between the three instruments. In addition to the good consistency of TROPOMI BrO retrievals with other satellite products, TROPOMI shows excellent performances with much smaller fitting

RMS values and lower random scatter of BrO columns than OMI and GOME-2B. More small-scale hotspots can be identified in greater detail by TROPOMI with its improved signal-to-noise ratio and the excellent spatial resolution. Thus, studies on small-scale BrO events in specific source regions where comparatively lower spatial resolution satellite sensors such as GOME, SCIAMACHY, GOME-2 and OMI provide only limited information and may even fail to detect the small-scale plume will be enabled by TROPOMI data.

In spite of the overall good performance of BrO retrievals on TROPOMI data, poor BrO spectral fits are sometimes found over inhomogeneous reflectance scenes such as fractional clouds and ice shelfs due to inhomogeneous slit illumination. This inhomogeneous scene effect on the DOAS retrieval should be corrected to obtain more accurate retrieval results. In this demonstration study, a simplified air mass factor and no stratospheric correction were applied because the main purpose of this study is to find the optimal DOAS retrieval settings for BrO that reduce systematic biases by minimizing effects of

interfering absorbers and to assess the consistency with previous satellite results. However, for future quantitative studies of tropospheric BrO explosion events, stratospheric correction and improved air mass factor calculation taking into account the observation conditions are essential. In particular, investigation and evaluation of high resolution input data applicable to the unprecedented small footprint of TROPOMI should be performed, which will be a subject of further work. In addition to the satellite intercomparisons shown here, validation with ground-based measurements is needed for more detailed assessment of

the quality of TROPOMI BrO columns. Judging from the examples evaluated in this study, BrO columns from TROPOMI will contribute relevant high resolution information to many future studies exploring the halogen chemistry in the atmosphere.

**Appendix A. Improvement of the BrO retrieval with the Pukite Taylor series approach**

To investigate the possibility of a DOAS fit improvement for the polar BrO measurement scenario by applying the Taylor series approach (Pukite et al., 2010), we performed an additional sensitivity test. The test was conducted in the same way and with the same measurement scenario as described in section 3.1.1, but two pseudo cross sections of $O_3$ at 223 K ($\lambda\sigma_{O3}$ and $\sigma_{O3}^2$) were added to the standard DOAS settings. The reason for choosing the lower temperature $O_3$ cross section is that this temperature is closer to the polar lower stratospheric temperature in spring. These two fitting parameters are terms derived by a Taylor series expansion to account for the wavelength dependency of the SCD which results from changes in light path distribution with wavelength and absorption strength (Pukite et al., 2010). Pukite et al. (2010) demonstrated that the application of the Taylor series approach to strong absorber $O_3$ leads to an improvement for the fit of the weaker absorber BrO in the UV range of limb measurements.

Fig. A1 shows BrO retrieval results obtained with the DOAS settings including the Taylor series approach for the TROPOMI polar BrO measurement scenario. Compared with the standard retrieval results (Fig. 2 in section 3.1.1), BrO retrieval results applying the Taylor series approach show reduced fitting RMS values across the whole retrieval wavelength range (see Fig. A2, right plot). In particular, fitting results at wavelength ranges with a start limit between 323-327.6 nm where negative BrO SCDs and high fitting errors occurred due to strong $O_3$ interference are significantly improved as BrO SCDs increased by ~$1.4\times10^{14}$ molec cm$^{-2}$ and fitting errors decreased by ~32 %. Also, the abrupt changes of BrO SCDs around 333 nm of start wavelength and wavelength range with start limits of 335-337.6 nm and end limits of 349-353.6 nm are moderated by use of the Taylor series expansion for $O_3$. These sensitivity test results using TROPOMI nadir measurements clearly demonstrate that introducing the Taylor series approach for $O_3$ results in an improvement of the DOAS fit. However, as is also clear from Figure A1, not all of the problems at low wavelengths apparent in Figure 2 are solved by including the Pukite terms.

For the fitting window selected in this study (334.6-358 nm), the application of the Taylor series approach for $O_3$ does not significantly affect BrO retrieval results compared with the standard DOAS retrieval. However, as can be seen from Fig. A2, effects of the Taylor series expansion for $O_3$ on the BrO SCD retrieval vary depending on the retrieval wavelength interval. The strength of absorption and the slant path of scattered light in the atmosphere vary considerably with wavelength, and thus the degree of improvement by the Taylor series approach for $O_3$ in the BrO retrieval is also different depending on the fitting wavelength range. Therefore, it is necessary to evaluate the improvement of the SCD retrieval by the Taylor series approach with respect to standard DOAS retrieval according to the fitting window selected. Moreover, we showed only sensitivity test results applying the Taylor series expansion of the lower temperature $O_3$ cross section to TROPOMI polar BrO measurements in this section, but note that the effect of the Taylor series approach may be different for different trace gas cross sections, temperature, and measurement scenarios.

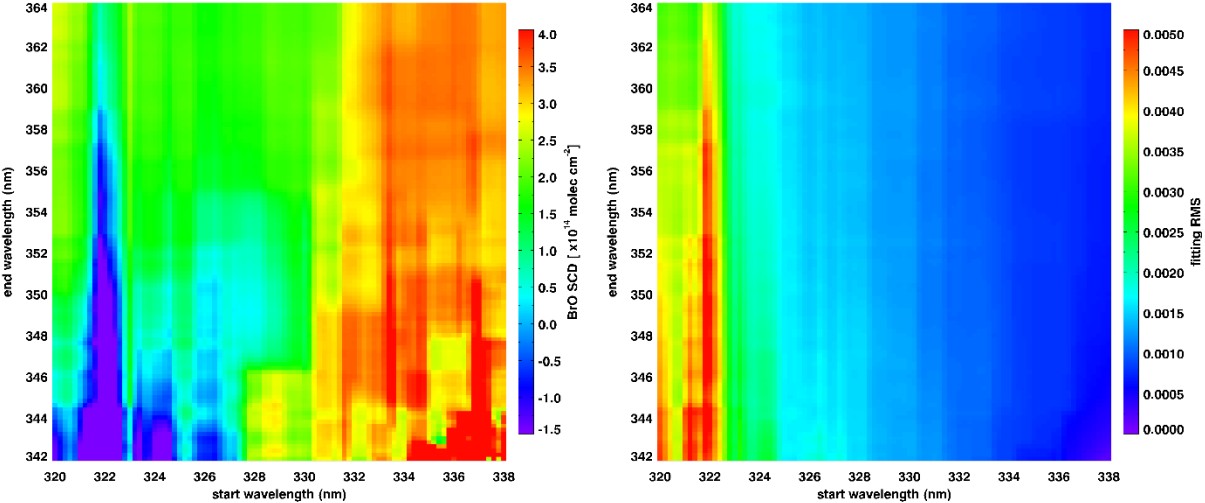

**Figure A1.** Color coded means of BrO SCDs (left) and fitting RMS values (right) retrieved when including the Taylor series approach for $O_3$ in the DOAS analysis.

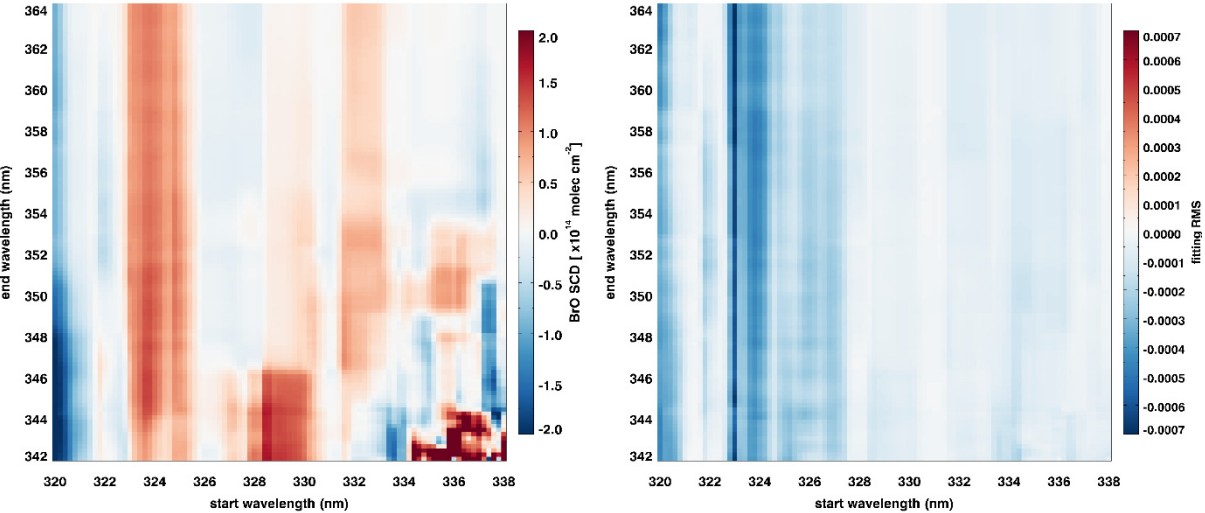

**Figure A2.** Color coded means of differences for BrO SCDs (left) and fitting RMS values (right) between analyses including the Taylor series approach (see Fig. A1) and the standard DOAS (see Fig.2 in section 3.1.1).

**Acknowledgements**

Parts of this study were funded through the University of Bremen, the DLR project 50EE1618 and the SFB/TR 172 "ArctiC Amplification: Climate Relevant Atmospheric and SurfaCe Processes, and Feedback Mechanisms (AC)³" in sub-project C03 by the DFG. GOME-2 lv1 data were provided by EMETSAT, OMI lv1 and lv2 data by NASA. Copernicus Sentinel 5P lv1 data from 2018 were used in this study. Sentinel-5 Precursor is a European Space Agency (ESA) mission implemented on behalf of the European Commission (EC). The TROPOMI payload is a joint development by ESA and the Netherlands Space

Office (NSO). The Sentinel-5 Precursor ground segment development has been funded by ESA and with national contributions from the Netherlands, Germany, Belgium and UK.

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
