# Peer review of "First high resolution BrO column retrievals from TROPOMI"

_Atmospheric Measurement Techniques, 2018_

## Referee Comment (RC1) · Anonymous Referee #2 · 25 Feb 2019

Seo et al. present first BrO data from TROPOMI, made a number of tests for optimal SCDs retrievals and present a number of interesting case studies. It is an interesting work. The manuscript is clearly written and the methodology is sound. This study should be published in Atmos. Meas. Tech. if the authors successfully address the following (minor) comments:

-the main remark is that section 3.1 on sensitivity test is rather long and ends up with findings that are mostly know. The method of Vogel is meaningful because it was originally applied on synthetic spectra. Here the technique is applied on real spectra, hence no firm conclusions can be drawn on the optimal wavelength range. The interference with O3 and SO2 at short UV is well known and the Ring effect at longer wavelength is not surprising either. The final selection of fitting interval of 333.5-357nm is not very

different from other studies and I encourage the authors to cite the past papers.

In section 3.1.1, it is written that additional cross-sections for ozone (Pukite et al., 2010) could improve the fits at shorter wavelengths. I propose to test this (simple) approach as it might further stabilize the retrievals.

References are missing: on page 1, studies on the presence of global BrO background in the free-troposphere should be listed. References for BrO observations by CIMS and long-path DOAS should be added. An important paper on MAXDOAS BrO in polar region is also: Frieß, U., H. Sihler, R. Sander, D. Pöhler, S. Yilmaz, and U. Platt (2011), The vertical distribution of BrO and aerosols in the Arctic: Measurements by active and passive differential optical absorption spectroscopy, J. Geophys. Res., 116, D00R04, doi:10.1029/2011JD015938. Finally the work of Theys et al., ACP, 2011 is absent throughout the manuscript and should be added.

Needs for clarifications:

Section 2: band 3 is not starting at 320 nm.

Page 3 l27: it is not only noise that results from interferences but also important are biases. Please clarify.

Page3 l30-31 is unclear. Please rephrase.

Table 1: for volcanic plume, the number of pixels is 1748. Is there not a mistake? It seems a lot, especially that it is mentioned in section 3.1.2 that it a 'small-scale BrO plume'. Please check.

Page 7, l16: It is speculated about the impact of the Ring effect due aerosol loads and cloud formation after the eruption. Is there any indication about this? Is the TROPOMI AAI product suggesting the presence of aerosols.

Section 3.2: a figure illustrating the offset correction would be nice. The approach is not very clear from the text (from l18).

P12, l28: The shift of OMI compared to other satellites is attributed to the 'relatively high measurement noise'. It is unclear. Fitting residuals from OMI are fine.

Figures 9 and 10 are difficult to read. Coastlines are not always visible. It would be good to improve the figures.

Figure 11: the MODIS pictures are shown but not really commented. What should be seen?

Conclusions, P19, l9: please add the obtained values for the slopes of the regression lines, in addition to the correlation coefficients.

———————————————————

---

## Referee Comment (RC2) · Anonymous Referee #1 · 7 Mar 2019

This is a well-written and detailed manuscript describing initial spectral fitting results from the TROPOMI sensor on Sentinel 5P. The images in the manuscript show dramatic improvement in identifying small-scale features that are allowed by the higher resolution of this new satellite sensor. The authors nicely describe their methods and selections that were made to avoid biases by other absorbers. I feel that the manuscript is appropriate for AMT, but have a few comments that I think should be addressed in a revised version.

Broader comments:

The authors show that formaldehyde causes a particularly challenging species to retrieve along with BrO, particularly when using narrow spectral windows. In the end the authors make what appears to be a good compromise to avoid the larger absorbers

and other issues, but one really wants to quantify all absorbers and not simply try to avoid them and then hope that the selection was good enough. Therefore, I think the manuscript should indicate that this window is a good choice, but future work needs to be done to continue to improve quantification of all the absorbers that occur. This window is a good start, but quantitative results will be improved by further research, which should be encouraged in the manuscript and conclusions.

The manuscript shows dramatic images of small scale BrO features that are amazing to see and will increase our scientific understanding. However, it is important that the manuscript points out that albedo effects are really important for quantification of lower-tropospheric BrO. Therefore, some things that are evident in the images using the simple geometric airmass factor should be pointed out as likely due to albedo effects. The reason for this is that I worry that a reader less aware of these issues may not take proper care of the albedo and misinterpret the images. An example that I think is likely an albedo effect is the sharp edge of small BrO evident North of Scandinavia/Russia in Figure 9a. That feature could be caused by the sea ice edge (http://nsidc.org/arcticseaicenews/2018/04/), with the high albedo of snow/sea ice allowing you to observe tropospheric BrO, while the low albedo of open ocean tends to prevent observation of lower tropospheric BrO. These albedo effects may also be important in some of the Rann of Kutch observations shown in Figure 13. The authors should point these effects out and indicate that they need to be taken into account for quantitative work.

The comparison of these retrievals to those of OMI and GOME-2B assist in assuring consistency between these different satellite sensors. However, they do not assure "accuracy" of the measurements; they show consistency. Therefore, on page 15, line 7, it should say "To evaluate the consistency of TROPOMI BrO..." Additionally, on page page 2, lines 26-28, the text should read "TROPOMI BrO columns were compared with those from the two existing satellite instruments, GOME-2B and OMI, with the consistency of these data sets were investigated." These comparisons do not either

"validate" this method, nor do they assure "accuracy". They are more "precise" and "consistent", which can be pointed out and are nice advances in the field.

I believe from the text on page 15, line 8 that the authors have used spectra from OMI and GOME-2B and retrieved BrO using the same processing methods applied here for TROPOMI data to compare with their TROPOMI measurements. Both OMI and GOME-2B have operational products, but my reading of this sentence is that those operational products were not used in this consistency check, but a single retrieval method (described here and using Table 2 settings) was used on spectral data from the three satellites. It needs to be made more clear whether spectra (Level 1) data or slant column (Level 2) operational products are being used. The reason it needs to be made more clear is that one might read that this paper says there is good correlation between this method and OMI, and take that to mean that the OMI operational product (OMBRO) agrees with this product, which is not what I think is plotted in Figure 10 panels. Alternatively, if the operational OMI and GOME-2B products are being used, they don't appear to be cited appropriately. Any references in the abstract and/or conclusions also needs to make clear what this "consistency" check is.

Specific comments:

Page 1, line 18: This should say "...reactions that deplete ozone..." (replace "which" with "that").

Page 2, line 27: Remove "verified" and "accuracy". Clarify if you are using operational products or re-analysis of spectra.

Page 3, line 2: Should say either "... which allows it to monitor..." or "... which allows monitoring of ..."

Page 10, line 15: The word "dependency" should be "dependence"

Page 12, line 2: Although this is a "robust selection", it would be valuable to point out that further work is warranted, particularly to retrieve both HCHO and BrO.
Page 17, Figure 11: As a courtesy to the reader, please point out the geographic region where this coastal edge occurs. I think it is the Kara Sea in the Russian Arctic. Potentially pointing out some island or landmass would help.

Page 18, Figure 12: Again, it would be nice to see where this image is – I think it is the Beaufort Sea, North of Alaska.

Page 20, Line 26: I is possible that using wavelengths longer than 362 nm had poorer results because of poor fitting of O4. O4's spectrum is temperature dependent and a warm temperature was chosen. If you cut the wavelength region at 362nm, then the polynomial can remove some aspects of the O4 absorption, while if you go beyond the peak, it gets harder for the polynomial to fit the features.

---

## Author Comment (AC1) · 4 Apr 2019

**Response to anonymous referee #1**

We thank the reviewers for their constructive comments and suggestions which helped us to improve our manuscript. We have addressed their questions as follows:

This is a well-written and detailed manuscript describing initial spectral fitting results from the TROPOMI sensor on Sentinel 5P. The images in the manuscript show dramatic improvement in identifying small-scale features that are allowed by the higher resolution of this new satellite sensor. The authors nicely describe their methods and selections that were made to avoid biases by other absorbers. I feel that the manuscript is appropriate for AMT, but have a few comments that I think should be addressed in a revised version.

Broader comments:

The authors show that formaldehyde causes a particularly challenging species to retrieve along with BrO, particularly when using narrow spectral windows. In the end the authors make what appears to be a good compromise to avoid the larger absorbers and other issues, but one really wants to quantify all absorbers and not simply try to avoid them and then hope that the selection was good enough. Therefore, I think the manuscript should indicate that this window is a good choice, but future work needs to be done to continue to improve quantification of all the absorbers that occur. This window is a good start, but quantitative results will be improved by further research, which should be encouraged in the manuscript and conclusions.

We have added a few more sentences that emphasize the need of further studies on the optimum fitting window and DOAS settings to solve the problems presented in this study as you suggested:

In section 3.1.6 (P12 L9 in the revised manuscript):

*Although the choice of the optimal fitting window may seem arbitrary to some degree, the analysis of several different relevant scenarios for many possible combinations of fitting windows described above demonstrates that this is an overall robust selection.*

{+ "However, further studies are needed to address the remaining challenges identified through the sensitivity tests, in particular the possible spectral cross correlation of BrO with HCHO around the selected fitting window."}

In conclusion (P24 L23):

*Based on the information gathered from the sensitivity tests for different measurement scenarios, 334.6-358 nm was selected as the optimal wavelength range for TROPOMI BrO retrievals for most of the possible measurement situations.*

{+ "This fitting window yields reliable BrO retrieval results with small fitting errors, but future studies on quantitative assessments and cross correlations between BrO and all the interfering absorbers are encouraged to further improve retrieval results."}

The manuscript shows dramatic images of small scale BrO features that are amazing to see and will increase our scientific understanding. However, it is important that the manuscript

points out that albedo effects are really important for quantification of lower-tropospheric BrO. Therefore, some things that are evident in the images using the simple geometric air mass factor should be pointed out as likely due to albedo effects. The reason for this is that I worry that a reader less aware of these issues may not take proper care of the albedo and misinterpret the images. An example that I think is likely an albedo effect is the sharp edge of small BrO evident North of Scandinavia/Russia in Figure 9a. That feature could be caused by the sea ice edge (http://nsidc.org/arcticseaicenews/2018/04/), with the high albedo of snow/sea ice allowing you to observe tropospheric BrO, while the low albedo of open ocean tends to prevent observation of lower tropospheric BrO. These albedo effects may also be important in some of the Rann of Kutch observations shown in Figure 13. The authors should point these effects out and indicate that they need to be taken into account for quantitative work.

We agree with the reviewer and have revised and added more text on uncertainties and biases of retrieved BrO vertical columns applying geometric AMFs in several parts of the manuscript:

In section 4.1 (P15 L4):

"The number of photons detected at the satellite is larger over bright surface areas than over dark surface areas. Therefore, the use of a simple AMF which does not consider the sensitivity to surface albedo can underestimate BrO vertical columns over dark surfaces such as the ocean in comparison to high surface albedo regions such as the Antarctic region, north of Russia and Canada. In addition to the surface albedo, clouds also affect signals detected at the satellite. The light path length and intensity are significantly changed by the cloud top height, cloud thickness and cloud fraction. Using an AMF that does not take into account the cloud effects can therefore result in errors in the computed vertical columns, as can be seen from the slightly higher BrO VCDs in the subpolar regions where cyclones are frequently observed due to the sub-polar low pressure system (Fig.11). Consequently, an improved AMF reflecting the sensitivity of surface albedo, cloud properties and BrO vertical profile should be calculated to obtain more accurate vertical column densities (Theys et al., 2011; Sihler et al., 2012), and this will be investigated in detail in a follow-up study using surface albedo and cloud information from the operational satellite products as they become available."

In section 4.3.1 (P20 L10):

"Convective clouds can be formed around open leads due to the supply of water vapor and enhanced vertical mixing, but computed BrO enhancement over clouds may have an error because of the use of AMFs which do not consider the effects of clouds."

In section 4.3.2 (P22 L16):

"It should be noted that the AMFs used in this work do not consider surface albedo and cloud effects, and therefore BrO VCDs may be overestimated over the bright salt marsh."

In conclusion (P25 L20):

"In this demonstration study, a simplified air mass factor and no stratospheric correction were applied because the main purpose of this study is to find the optimal DOAS retrieval settings for BrO that reduce systematic biases by minimizing effects of interfering absorbers and to assess the consistency with previous satellite results. However, for future quantitative

studies of tropospheric BrO explosion events, stratospheric correction and improved air mass factor calculation taking into account the observation conditions are essential. In particular, investigation and evaluation of high resolution input data applicable to the unprecedented small footprint of TROPOMI should be performed, which will be a subject of further work."

The comparison of these retrievals to those of OMI and GOME-2B assist in assuring consistency between these different satellite sensors. However, they do not assure "accuracy" of the measurements; they show consistency. Therefore, on page 15, line 7, it should say "To evaluate the consistency of TROPOMI BrO..." Additionally, on page page 2, lines 26-28, the text should read "TROPOMI BrO columns were compared with those from the two existing satellite instruments, GOME-2B and OMI, with the consistency of these data sets were investigated." These comparisons do not either "validate" this method, nor do they assure "accuracy". They are more "precise" and "consistent", which can be pointed out and are nice advances in the field.

We agree with your comment on the problem of "accuracy" word usage because we didn't validate our retrieval results using independent observations but identified consistencies of BrO retrievals between TROPOMI and previous satellites. Thus, the term "accuracy" has been removed in the revised manuscript as you suggested.

P2 L27: TROPOMI BrO columns were verified -> TROPOMI BrO columns were assessed

P2 L28: with their accuracy -> removed

P17 L7: To evaluate the accuracy and consistency -> To evaluate the consistency

I believe from the text on page 15, line 8 that the authors have used spectra from OMI and GOME-2B and retrieved BrO using the same processing methods applied here for TROPOMI data to compare with their TROPOMI measurements. Both OMI and GOME-2B have operational products, but my reading of this sentence is that those operational products were not used in this consistency check, but a single retrieval method (described here and using Table 2 settings) was used on spectral data from the three satellites. It needs to be made more clear whether spectra (Level 1) data or slant column (Level 2) operational products are being used. The reason it needs to be made more clear is that one might read that this paper says there is good correlation between this method and OMI, and take that to mean that the OMI operational product (OMBRO) agrees with this product, which is not what I think is plotted in Figure 10 panels. Alternatively, if the operational OMI and GOME-2B products are being used, they don't appear to be cited appropriately. Any references in the abstract and/or conclusions also needs to make clear what this "consistency" check is.

We have revised the text to make clear the data used in the intercomparison of satellite retrievals as:

P14 L14: "Applying the retrieval settings described in Table 2, BrO vertical columns have been computed from TROPOMI, OMI and GOME-2B Level 1 spectra."

P17 L8: "To evaluate the consistency of TROPOMI BrO retrievals with those from other satellites, a comparison was performed using GOME-2B and OMI retrievals obtained by

applying the same retrieval setting (Table 2) to Level 1b data."

Specific comments:

Page 1, line 18: This should say "...reactions that deplete ozone..." (replace "which" with "that").

This has been corrected in the revised manuscript.

Page 2, line 27: Remove "verified" and "accuracy". Clarify if you are using operational products or re-analysis of spectra.

Thanks for pointing this out. We have removed the expressions "verified" and "accuracy" in the revised manuscript.

P2 L27 TROPOMI BrO columns were verified -> TROPOMI BrO columns were assessed

P2 L28: with their accuracy -> removed

P17 L7: To evaluate the  consistency -> To evaluate the consistency

Page 3, line 2: Should say either "... which allows it to monitor..." or "... which allows monitoring of ..."

This has been corrected in the revised manuscript.

Page 10, line 15: The word "dependency" should be "dependence"

This has been corrected in the revised manuscript.

Page 12, line 2: Although this is a "robust selection", it would be valuable to point out that further work is warranted, particularly to retrieve both HCHO and BrO.

We have added more text to emphasize the need for further studies in section 3.1.6 as:

"However, further studies are needed for the remaining challenges identified through the sensitivity tests, in particular the possible spectral cross correlation of BrO with HCHO around the selected fitting window."

Page 17, Figure 11: As a courtesy to the reader, please point out the geographic region where this coastal edge occurs. I think it is the Kara Sea in the Russian Arctic. Potentially pointing out some island or landmass would help.

Figures have been modified to high resolution with adding texts denoting the region in the revised manuscript.

- Figure 13 in the revised manuscript:

[Figure]

Page 18, Figure 12: Again, it would be nice to see where this image is – I think it is the Beaufort Sea, North of Alaska.

Figures have been modified to high resolution with adding texts denoting the region in the revised manuscript.

- Figure 14 in the revised manuscript:

[Figure]

Page 20, Line 26: It is possible that using wavelengths longer than 362 nm had poorer results because of poor fitting of O4. O4's spectrum is temperature dependent and a warm temperature was chosen. If you cut the wavelength region at 362 nm, then the polynomial can remove some aspects of the O4 absorption, while if you go beyond the peak, it gets harder for the polynomial to fit the features.

Thanks for pointing this out. We have further mentioned potential problem of O4 fitting.

In section 3.1.6 (P11 L23):

"In the case of DOAS retrievals over the cloudy background region and the volcanic plume, higher fitting errors were found in the wavelength intervals extending beyond 362 nm because of imperfect correction of the Ring effect and possibly also poorer fitting of $O_4$ related to the temperature dependency of the cross section."

In conclusion (P24 L20):

"At wavelength longer than 362 nm, poorer results were found in the cloudy scene and volcanic plume measurement scenario, presumably due to the wavelength dependency of the Ring effect and imperfect fitting of $O_4$."

---

## Author Comment (AC2) · 4 Apr 2019

**Response to anonymous referee #2**

We thank the reviewers for their constructive comments and suggestions which helped us to improve our manuscript. We have addressed their questions as follows:

Seo et al. present first BrO data from TROPOMI, made a number of tests for optimal SCDs retrievals and present a number of interesting case studies. It is an interesting work. The manuscript is clearly written and the methodology is sound. This study should be published in Atmos. Meas. Tech. if the authors successfully address the following (minor) comments:

- The main remark is that section 3.1 on sensitivity test is rather long and ends up with findings that are mostly know. The method of Vogel is meaningful because it was originally applied on synthetic spectra. Here the technique is applied on real spectra, hence no firm conclusions can be drawn on the optimal wavelength range. The interference with O3 and SO2 at short UV is well known and the Ring effect at longer wavelength is not surprising either. The final selection of fitting interval of 333.5-357nm is not very different from other studies and I encourage the authors to cite the past papers.

We agree that the results of the sensitivity study are not very surprising considering results from previous work on synthetic data and other measurements. However, we disagree that the sensitivity tests on real data are less relevant than the synthetic data tests from Vogel et al. While the fact that we do not know the true column is a limitation, the use of real data from different realistic scenarios makes the study much more relevant for real data analysis. Synthetic data are great to investigate the theoretical limitations of a technique like DOAS, but many real world problems are not fully reflected by studies on synthetic data.

Following the suggestions for the reviewer, more references have been included in the text where appropriate.

In section 3.1.1:

Aliwell, S. R., Van Roozendael, M., Johnston, P. V., Richter, A., Wagner, T., Arlander, D. W., Burrows, J. P., Fish, D. J., Jones, R. L., Tornkvist, K. K., Lambert, J. C., Pfeilsticker, K., and Pundt, I.: Analysis for BrO in zenith-sky spectra: An intercomparison exercise for analysis improvement, Journal of Geophysical Research-Atmospheres, 107, 10.1029/2001jd000329, 2002.

Theys, N., Van Roozendael, M., Hendrick, F., Yang, X., De Smedt, I., Richter, A., Begoin, M., Errera, Q., Johnston, P. V., Kreher, K., and De Maziere, M.: Global observations of tropospheric BrO columns using GOME-2 satellite data, Atmospheric Chemistry and Physics, 11, 1791-1811, 10.5194/acp-11-1791-2011, 2011.

In section 3.1.3:

Theys, N., Van Roozendael, M., Dils, B., Hendrick, F., Hao, N., and De Mazière, M.: First satellite detection of volcanic bromine monoxide emission after the Kasatochi eruption, Geophysical Research Letters, 36.3, 10.1029/2008gl036552, 2009.

In section 3.1.5:

Burrows, J. P., Platt, U., and Borrell, P.: The remote sensing of tropospheric composition

from space, Springer Science & Business Media, 2011.

Theys, N., Van Roozendael, M., Hendrick, F., Yang, X., De Smedt, I., Richter, A., Begoin, M., Errera, Q., Johnston, P. V., Kreher, K., and De Maziere, M.: Global observations of tropospheric BrO columns using GOME-2 satellite data, Atmospheric Chemistry and Physics, 11, 1791-1811, 10.5194/acp-11-1791-2011, 2011.

In section 3.1.1, it is written that additional cross-sections for ozone (Pukite et al., 2010) could improve the fits at shorter wavelengths. I propose to test this (simple) approach as it might further stabilize the retrievals.

As you suggested, we performed additional sensitivity test for the polar BrO measurement scenario by adding 2 Pukite pseudo cross sections of O3 at 223 K to the standard DOAS setting. The sensitivity test results show a clear improvement for the fit in particular in the shorter wavelength range. We have added test results and their analysis in Appendix A:

[revised manuscript text omitted]

References are missing: on page 1, studies on the presence of global BrO background in the free-troposphere should be listed. References for BrO observations by CIMS and long-path DOAS should be added. An important paper on MAXDOAS BrO in polar region is also: Frie? U., H. Sihler, R. Sander, D. Pöhler, S. Yilmaz, and U. Platt (2011), The vertical distribution of BrO and aerosols in the Arctic: Measurements by active and passive differential optical absorption spectroscopy, J. Geophys. Res., 116, D00R04, doi:10.1029/2011JD015938. Finally the work of Theys et al., ACP, 2011 is absent throughout the manuscript and should be added.

As you suggested, references have been added in the revised manuscript:

P1 L26 (in the revised manuscript): "BrO observations have been carried out by in-situ chemical ionization mass spectrometry (CIMS) (Liao et al., 2011; Choi et al., 2012), ground-based differential optical absorption spectroscopy (DOAS) measurements such as long-path DOAS (LP-DOAS) (Hönninger et al., 2004; Liao et al., 2011; Stutz et al., 2011) and multi-axis DOAS (MAX-DOAS) (Hönninger et al., 2004; Frieß et al., 2011; Zhao et al., 2016)."

P2 L14: "The higher spatial resolution data of GOME-2 and OMI have been successfully used to monitor daily global distribution (Theys et al., 2011) ~

P6 L29: "a spectral cross correlation between BrO and HCHO were also identified in Theys et al. (2011) and Vogel et al. (2013)"

Liao, J., Sihler, H., Huey, L., Neuman, J., Tanner, D., Friess, U., Platt, U., Flocke, F. M., Orlando, J. J., Shepson, P. B., Beine, H. J., Weinheimer, A. J., Sjostedt, S. J., Nowak, J. B., Knapp, D. J., Staebler, R. M., Zheng, W., Sander, R., Hall, S. R., and Ullmann, K.: A comparison of Arctic BrO measurements by chemical ionization mass spectrometry and long path-differential optical absorption spectroscopy, J. Geophys. Res., 116, D00R02, doi:10.1029/2010JD014788, 2011.

Stutz, J., Thomas, J. L., Hurlock, S. C., Schneider, M., von Glasow, R., Piot, M., Gorham, K., Burkhart, J. F., Ziemba, L., Dibb, J. E., and Lefer, B. L.: Longpath DOAS observations of surface BrO at Summit, Greenland, Atmospheric Chemistry and Physics, 11, 9899-9910, 10.5194/acp-11-9899-2011, 2011.

Hönninger, G., Leser, H., Sebasti´an, O., and Platt, U.: Ground-based measurements of halogen oxides at the Hudson Bay by active longpath DOAS and passive MAX-DOAS, Geophys. Res. Lett., 31, L04111, doi:10.1029/2003GL018982, 2004.

Frieß, U., H. Sihler, R. Sander, D. Pöhler, S. Yilmaz, and U. Platt (2011), The vertical distribution of BrO and aerosols in the Arctic: Measurements by active and passive differential optical absorption spectroscopy, J. Geophys. Res., 116, D00R04, doi:10.1029/2011JD015938.

Zhao, X., Strong, K., Adams, C., Schofield, R., Yang, X., Richter, A., Friess, U., Blechschmidt, A. M., and Koo, J. H.: A case study of a transported bromine explosion event in the Canadian high arctic, Journal of Geophysical Research-Atmospheres, 121, 457-477, 10.1002/2015jd023711, 2016.

Theys, N., Van Roozendael, M., Hendrick, F., Yang, X., De Smedt, I., Richter, A., Begoin, M., Errera, Q., Johnston, P. V., Kreher, K., and De Maziere, M.: Global observations of tropospheric BrO columns using GOME-2 satellite data, Atmospheric Chemistry and Physics, 11, 1791-1811, 10.5194/acp-11-1791-2011, 2011.

Needs for clarifications:

Section 2: band 3 is not starting at 320 nm.

According to the latest version of TROPOMI L01b ATBD and IODS, the spectral performance range of band 3 is specified as 320-405 nm. While there is data at shorter wavelengths, it is in the overlapping regions between band 2 and band 3.

[1] Table 1 of P 22, ATBD; Algorithm theoretical basis document for the TROPOMI L01b data processor; source: KNMI, ref: S5P-KNMI-L01B-0009-SD; issue: 8.0.0; date: 2017-06-01;

url: https://sentinel.esa.int/documents/247904/2476257/Sentinel-5P-TROPOMI-Level-1B-ATBD

[2] Table 1 of P 15, IODS; Input/output data specification for the TROPOMI L01b data processor; source: KNMI; ref: S5P-KNMI-L01B-0012-SD; issue: 9.0.0; date: 2018-04-01;

url: https://sentinel.esa.int/documents/247904/3119978/Sentinel-5P-Level-01B-input-output-data-specification

Veefkind et al. (2012) specified the spectral range of band 3 as 310-405 nm, but this paper was written before the launch of TROPOMI and contains outdated information that is different from the current state (ex. the current spatial sampling for UV/vis band is 3.5x7 $km^2$, not 7x7 $km^2$; the current spectral performance range of band 3 is 320-405 nm, not 310-405 nm).

Page 3 l27: it is not only noise that results from interferences but also important are biases. Please clarify.

The sentence has been changed to as (P3 L29 in the revised manuscript):

"In general, larger fitting windows can improve the quality of DOAS retrievals by using more spectral points, but at the same time, they can increase the noise and bias resulting from interfering signals with other absorbers and wavelength dependent light path lengths."

Page3 l30-31 is unclear. Please rephrase.

The sentence has been rephrased as (P4 L4 in the revised manuscript):

"Thus, finding a compromise for a fitting window that avoids the disadvantages as well as making the best use of the advantages from the retrieval wavelength interval is important to yield the best quality DOAS retrieval result."

Table 1: for volcanic plume, the number of pixels is 1748. Is there not a mistake? It seems a lot, especially that it is mentioned in section 3.1.2 that it a 'small-scale BrO plume'. Please check.

We have checked the number of s5p satellite pixel used for each scenario test. The number of pixels used in volcanic BrO sensitivity test is 1748 and there was no mistake (You can identify the domain of the volcanic plume scenario in Figure 6). However, for the salt marsh sensitivity test, 137 pixels instead of 113 were used and this has been corrected in Table 1.

Page 7, l16: It is speculated about the impact of the Ring effect due aerosol loads and cloud formation after the eruption. Is there any indication about this? Is the TROPOMI AAI product suggesting the presence of aerosols?

As you suggested, we have added this figure in section 3.1.3:

[Figure]

Figure 6. TROPOMI UV aerosol index (340 nm/380 nm) from the operational Level 2 product and OMI SO$_2$ vertical columns [DU] from the column amount SO$_2$ TRM (mid-troposphere) of the operational OMSO2 product for a volcanic BrO measurement scenario. The domain used for the sensitivity test is indicated by a gray dashed box.

Section 3.2: a figure illustrating the offset correction would be nice. The approach is not very clear from the text (from l18).

As you suggested, we have added this figure in section 3.2:

[Figure]

Figure 10. Illustration showing destriping and offset correction steps described in section 3.2 using TROPOMI orbit 2207 on Mar 17 2018. BrO SCDs retrieved by daily row-dependent mean radiances in the Pacific reference sector as background spectrum for the across-track correction (left), offset-corrected BrO SCDs treated by applying the normalization approach including the VZA dependency on the BrO SCDs (middle), BrO VCDs computed by dividing the offset-corrected BrO SCDs by geometric AMFs (right).

P12, l28: The shift of OMI compared to other satellites is attributed to the 'relatively high measurement noise'. It is unclear. Fitting residuals from OMI are fine.

Compared with GOME-2B and TROPOMI launched in 2012 and 2017, OMI launched in 2004 shows the most severe degradation among the three satellites. The quality of OMI Level 1b radiance data is affected by the row anomaly, which causes errors in the BrO retrieval from the L1b spectra. Although OMI pixels affected by row anomaly were not used in the intercomparison study, the lower spectral stability of OMI compared to the other two satellites leads to systematic errors and biases in the BrO retrieval, which can be confirmed by a slightly positive biased BrO distribution in the Pacific background.

We have revised the sentence as (P15 L22 in the revised manuscript):

"The latter is attributed to be a consequence of systematic biases caused by the relatively lower quality of Level 1b radiance due to the instrument degradation."

Figures 9 and 10 are difficult to read. Coastlines are not always visible. It would be good to improve the figures.

Figures have been modified to high resolution with adding texts denoting the region in the revised manuscript.

Figure 11 in the revised manuscript:

[Figure]

Figure 12 in the revised manuscript:

[Figure]

Figure 11: the MODIS pictures are shown but not really commented. What should be seen?

We have revised and added more text in the Section 4.3.1 as:

"Another example of a BrO explosion event case is shown in Figure 14. A relatively narrow and long shape of enhanced BrO over the Beaufort Sea can be found in all three satellite maps. As discussed for the previous example, TROPOMI data with the high spatial resolution of 3.5x7 $km^2$ yield a more detailed view of the BrO explosion event compared to OMI and GOME-2B. The enhanced BrO plumes appear around open leads and sea ice cracks shown as slightly darker areas in the matching MODIS image (arrows pointing at examples). In particular, the elevated BrO around the Banks Island and the eastern Beaufort Sea (-140~-120 °E, 70~77 °N) could be significantly linked to open leads since frost flowers and sea salt aerosols which act as the source of reactive bromine can be formed in such areas (Simpson et al., 2007). Also, opening of sea ice leads can locally create enhanced vertical mixing and uplifting of bromine sources. However, the analysis of the long enhanced BrO plume from the coast of Alaska towards north should be cautious. The MODIS image composed of the 7-2-1 bands can distinguish clouds (as white) from the sea ice (as sky blue), and this image shows that the shape of the enhanced BrO plume is similar to that of clouds. Convective clouds can be formed around open leads due to the supply of water vapor and enhanced vertical mixing, but computed BrO enhancement over clouds may have an error because of the use of AMFs which do not consider the effects of clouds. In spite of this uncertainty, the enhancement of vertical columns by up to 4x10$^{13}$ molec cm$^{-2}$ compared to the surrounding values indicates that open leads could be associated to the BrO enhancement. Small-scale BrO explosion events around open leads or polynyas can be better investigated with the high spatial resolution TROPOMI data and will be the topic of a follow-up study."

Conclusions, P19, l9: please add the obtained values for the slopes of the regression lines, in addition to the correlation coefficients.

We have added slope values of the regression lines as:

"TROPOMI BrO retrievals show good agreements with OMI and GOME-2B BrO columns with high correlation coefficients (slopes of the regression lines) of 0.84 (0.89) and 0.84 (0.72) for enhanced BrO plumes in Arctic sea ice region, respectively."